# Breaking the Gridlock: Efficient Atmospheric Data Reconstruction and Prediction via Generative 3D Gaussian Splatting

## Abstract

AI-based numerical weather prediction (NWP) models often rely on regular latitude–longitude grids that induce strong data redundancy, limiting scalability to higher resolutions and wasting computation. We present *GaussianCast*, a generative 3D Gaussian Splatting (3DGS) framework for compact, continuous representation and efficient forecasting of high-dimensional atmospheric fields. To reduce redundancy while preserving global consistency, we place Gaussian centers on a Reduced Gaussian Grid (RGG), achieving equal-area sampling and enabling up to 14× compression. Conditioned on the current atmospheric state, multi-scale Graph Attention Transformers generate 3DGS covariances, occupancy, and attributes for both reconstruction and forecasting. On ERA5 dataset, GaussianCast achieves accurate weathr reconstruction and skillful medium-range weather forecasting at substantially lower computational cost, and remains competitive on tropical cyclone tracks. To our knowledge, it is the first generative 3DGS NWP framework to place Gaussians on RGG and predict their parameters for reconstruction and forecasting. Code is available at: https://anonymous.4open.science/r/GaussianCast-9F7B.

## 1 Introduction

Weather forecasting is vital for economic activity, public safety, and disaster preparedness. Despite recent advances, Artificial Intelligence (AI)-based Numerical Weather Prediction (NWP) models Lam et al. (2023); Bi et al. (2022); Chen et al. (2023a); Kochkov et al. (2024) are still limited by their reliance on regular latitude–longitude grids, which introduce significant data redundancy, especially near the poles, resulting in inefficient computation and wasted resources. This redundancy hinders the scalability of models to higher resolutions, which are crucial for accurately predicting extreme weather events such as hurricanes, heatwaves, and heavy rainfall. However, achieving these higher resolutions on regular grids requires substantial computational resources, posing a major obstacle to advancing weather forecasting.

Inspired by recent advances in 3D Gaussian Splatting (3DGS) for 3D scene reconstruction Mildenhall et al. (2021); Kerbl et al. (2023); Huang et al. (2024b); Dong et al. (2025), we observe that discrete atmospheric grid data can be efficiently represented in the continuous domain using 3DGS, enabling substantial data compression. Based on this insight, we propose *GaussianCast*, a generative 3DGS framework for numerical weather reconstruction and forecasting that achieves up to 14× compression and significantly improves forecasting efficiency. To adapt 3DGS for atmospheric data, we address excessive point density near the poles by adopting the classical Reduced Gaussian Grid (RGG) strategy Hortal & Simmons (1991), as used by the European Centre for Medium-Range Weather Forecasts (ECMWF) Bauer et al. (2015) in its Integrated Forecast System (IFS), which transitions from redundant latitude-longitude grids to more efficient designs. This approach strategically reduces grid points, especially at high latitudes, while maintaining global physical

consistency. By modeling RGG grid points as Gaussian centers, we define the 3DGS parameters, including covariance matrices, occupancy, and attributes, to represent and forecast atmospheric fields. Importantly, we extend Gaussian attributes to encode all 160 atmospheric variables (e.g., temperature, humidity, wind speed), rather than conventional RGB, enabling high-dimensional weather field representation.

However, Most existing 3DGS methods overfit individual samples and lack the generalization needed to handle unseen instances Zhang et al. (2024a;b), limiting their effectiveness for accurate weather representation and prediction. To overcome this, we propose a generative 3DGS framework conditioned on the current atmospheric state, enabling both reconstruction of the current frame and prediction of future frames by generating the relevant parameters. Specifically, our approach employs multi-scale Graph Attention Transformers (GATs) to dynamically generate essential 3DGS parameters, including covariance matrices, attributes, and occupancy, conditioned on RGG locations and observed atmospheric variables. By avoiding redundant latitude–longitude grids, this architecture not only provides a continuous and compact representation of atmospheric data but also improves computational efficiency and reduces storage costs, enabling robust and flexible forecasting.

We conduct extensive experiments on the ERA5 reanalysis dataset Hersbach et al. (2020), focusing on the representation of up to 160 atmospheric variables (e.g., temperature, humidity, wind speed). Results show that our approach significantly reduces reconstruction errors while providing a compact and efficient representation of high-dimensional atmospheric data. Moreover, Our method efficiently generates Gaussian parameters for future frames in medium-range weather forecasting, achieving performance comparable to advanced models such as GraphCast Lam et al. (2023) and Pangu-Weather Bi et al. (2022), but with substantially lower computational cost. Beyond general atmospheric fields, we further apply our 3DGS-based representation and prediction framework to extreme weather events, such as typhoons, achieving competitive forecasting performance, which highlights its practical significance. Our contributions are summarized as follows:

• We introduce the first generative 3DGS-based framework that enables high-quality reconstruction and accurate prediction of high-dimensional atmospheric data over continuous distributions.

• We place 3DGS points on reduced Gaussian grids to exploit the spherical structure of atmospheric data, achieving equal-area sampling that cuts redundancy and compresses data by up to 14×.

• Conditioned on latitude–longitude grid inputs, our method reconstructs atmospheric fields with under 5% error and achieving competitive performance with state-of-the-art medium-range weather forecasting models.

## 2 RELATED WORK

**AI-Based Weather Forecasting**   Recent advancements in AI-based weather forecasting have significantly enhanced medium-range prediction capabilities. Early efforts include FourCastNet Pathak et al. (2022), which introduced adaptive Fourier neural operators for global high-resolution forecasts up to 7 days. Subsequently, Pangu-Weather Bi et al. (2022) employed 3D convolutional neural networks to deliver fast and accurate global forecasts covering 1 hour to 7 days, while GraphCast Lam et al. (2023) utilized graph neural networks to model spatial correlations, achieving skillful medium-range forecasts up to 10 days, outperforming ECMWF's High-Resolution Forecast (HRES) on over 90% of verification targets. Fengwu Chen et al. (2023a) extended global medium-range forecasts beyond 10 days, showcasing machine learning's potential for extended predictions. NeuralGCM Kochkov et al. (2024) introduced a neural general circulation model for medium-range forecasting, followed by GenCast Price et al. (2023), which enhanced predictions with diffusion-based ensemble forecasting and uncertainty quantification. FengWu-4DVar Xiao et al. (2023) and FengWu-Adas Chen et al. (2023b) integrated data assimilation techniques to explore the end-to-end medium-range weather forecasting. Fengwu-GHR Han et al. (2024) achieves the $0.1°$ kilometer-scale medium-range predictions with limited high-resolution data, and ExtremeCast Xu et al. (2024a) targets extreme weather events within 7 days. KARINA Cheon et al. (2024) achieves accurate global weather forecasting at $2.5°$ resolution with minimal resources. WeatherGFT Xu et al. (2024b) combines a PDE kernel and neural networks to generalize

weather forecasts to finer temporal scales beyond the training dataset. Aurora Bodnar et al. (2024) integrated multi-source data for enhanced accuracy, Prithvi WxC Schmude et al. (2024) supports diverse weather and climate tasks like forecasting and downscaling. Finally, AIFS Lang et al. (2024a), ArchesWeatherGen Couairon et al. (2024), GraphDOP Alexe et al. (2024), and AIFS-CRPS Lang et al. (2024b) from ECMWF combined AI with traditional NWP strengths for medium-range forecasting. However, these models rely on latitude–longitude grids, causing data redundancy and resource costs Reichstein et al. (2019); Brenowitz & Bretherton (2019). In contrast, we place 3DGS centers on Reduced Gaussian Grids (RGG), achieving a compact, continuous representation of atmospheric data with greater data compression.

**3D Gaussian Splatting**  3D Gaussian Splatting (3DGS), introduced for real-time radiance field rendering, represents point clouds as 3D Gaussian distributions parameterized by position, covariance, and opacity Kerbl et al. (2023). Its adaptive density control and differentiable rasterization enable efficient, high-quality rendering, surpassing Neural Radiance Fields (NeRFs) in speed and scalability for 3D scene reconstruction Mildenhall et al. (2021); Kerbl et al. (2023); Zhou et al. (2024); Cheng et al. (2024). 3DGS has been applied to tasks such as dynamic scene tracking and editable scene synthesis, leveraging its explicit Gaussian representations Luiten et al. (2024); Huang et al. (2024a). Recent extensions to 2D Gaussian Splatting have explored image representation and compression, where Gaussian distributions model pixel data with parameters like position, rotation, and scaling Zhang et al. (2024a;b). For instance, GaussianImage achieves high-fidelity image reconstruction at 1000 FPS, demonstrating the efficiency of Gaussian-based modeling for 2D data Zhang et al. (2024a). Despite these advances, Gaussian splatting faces significant generalization limitations. Our work is to propose a generative 3DGS framework that transforms 3DGS to a conditional generation task, enabling generalized 3DGS for rendering weather data.

## 3   WEATHER RECONSTRUCTION AND FORECASTING ON GAUSSIAN SPACE

### 3.1   ATMOSPHERIC DATA RECONSTRUCTION WITH 3DGS

**3DGS Initialization on Reduced Gaussian Grid.**  Originally developed for real-time radiance field rendering, 3DGS represents point clouds as collections 3D Gaussians Kerbl et al. (2023). Each Gaussian is defined by a position vector of the center $\mu \in \mathbb{R}^3$, a covariance matrix $\Sigma \in \mathbb{R}^{3 \times 3}$ controlling shape and orientation, an opacity factor $\alpha \in [0, 1)$ for rendering. We extend the 3DGS framework for representing atmospheric data. Specifically, we conceptualize the atmospheric field as a function $F : S^2 \to \mathbb{R}^{160}$, where $S^2$ represents the Earth's surface as a unit sphere, and $\mathbb{R}^{160}$ corresponds to 160 atmospheric variables, such as temperature, humidity, and wind speed (see Appendix for details). As shown in Figure 1 a), the RGG provides $K$ quasi-uniform grid points $\{p_i\}_{i=1}^K$ on $S^2$, reducing redundancy at high latitudes Hortal & Simmons (1991). Specifically, the RGG defines points by latitudes $\phi_k$ (where $k = 1, \ldots, N_{\text{lat}}$, derived from Gaussian quadrature points in $[-90°, 90°]$) and longitudes $\lambda_{k,m}$ (where $m = 1, \ldots, M_k$). The number of longitudes $M_k$ at each latitude $\phi_k$ is reduced at higher latitudes, approximated as $M_k \approx M_{\text{max}} \cdot \cos \phi_k$, where $M_{\text{max}}$ is the number of longitudes at the equator (i.e., at $\phi = 0$), ensuring a quasi-uniform grid distribution across the sphere. Each RGG point $p_i$ corresponds to a pair $(\phi_k, \lambda_{k,m})$. The atmospheric field is represented by a collection of 3D Gaussians $\mathcal{G} = \{\mathcal{G}_i\}_{i=1}^K$, where each Gaussian $\mathcal{G}_i = (\mu_i, \Sigma_i, f_i, \alpha_i)$ is defined by the probability density function:

$$\mathcal{G}_i(p) = \alpha_i \cdot \frac{1}{(2\pi)^{3/2} |\Sigma_i|^{1/2}} \exp\left(-\frac{1}{2}(p - \mu_i)^T \Sigma_i^{-1} (p - \mu_i)\right), \tag{1}$$

parameterized by its position $\mu_i \in \mathbb{R}^3$, a covariance matrix $\Sigma_i \in \mathbb{R}^{3 \times 3}$, a feature vector $f_i \in \mathbb{R}^{160}$ storing the 160 variable values at $p_i$, and an opacity factor $\alpha_i \in [0, 1)$. The position $\mu_i$ is derived by transforming the RGG point $p_i$ into a 3D Cartesian position, serving as the center of the Gaussian distribution: $\mu_i = (\cos \phi_k \cos \lambda_{k,m}, \cos \phi_k \sin \lambda_{k,m}, \sin \phi_k)$. The covariance is constructed as $\Sigma_i = RSS^T R^T$, where $R$

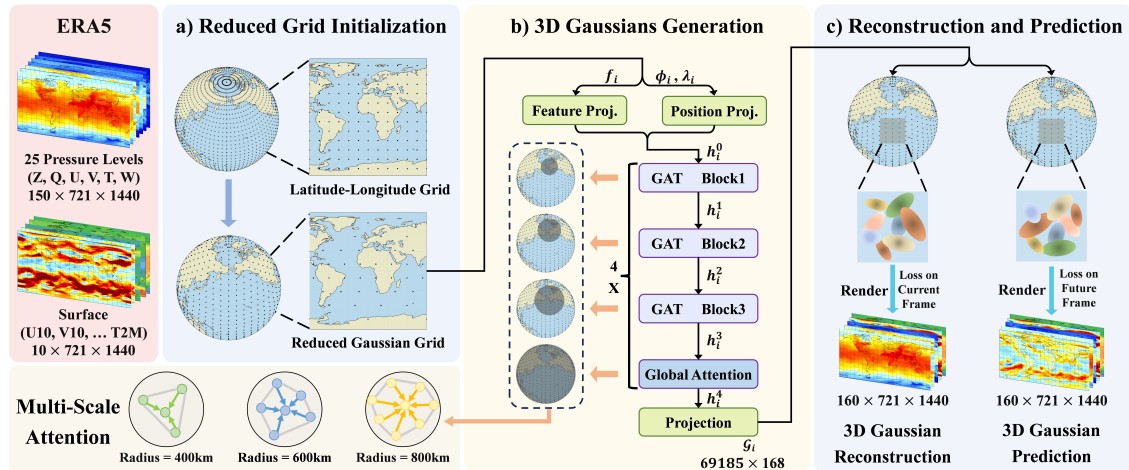

Figure 1: **3DGS-based atmospheric data reconstruction and prediction on RGG grids**. a) ERA5 (160×721×1440) is mapped onto RGGs to eliminate high-latitude redundancy. b) 3D Gaussians Prediction model utilizes multi-scale graph attention and global attention, capturing spatial correlations for generating 3D Gaussian parameters. c) Loss on current-frame data yields atmospheric representation and compression, reducing the original 160×721×1440 variables to 69,185×168 Gaussians (Compression Ratio ≈ 1/14), while loss on future-frame data enables atmospheric prediction.

is a rotation matrix from quaternion $q_i$ and $S = \mathrm{diag}(s_{i1}, s_{i2}, s_{i3})$ scales along three axes Kerbl et al. (2023). This allows each Gaussian to adapt its shape and orientation during optimization. The atmospheric field is thus represented by the collection $\mathcal{G} = \{\mathcal{G}_i\}_{i=1}^K$.

## 3.2 CONDITIONAL 3DGS GENERATION

**Problem Formulation.** Unlike existing AI forecasting models such as Pangu-Weather Bi et al. (2022) and GraphCast Lam et al. (2023), which predict directly on latitude-longitude grids, our method predicts a set of Gaussian distributions representing the atmospheric field. Instead of predicting future Gaussian distributions from the rendered Gaussian space at time $t$, we directly generate the Gaussian distributions at the next time step $t + 1$ using the raw atmospheric data at time $t$ as a conditional input. Specifically, as depicted in the Figure 1 b), given the ERA5 atmospheric field $F(t) : S^2 \to \mathbb{R}^{160}$ at time $t$, represented as a tensor of shape $160 \times 721 \times 1440$, our objective is to generate the Gaussian space $\mathcal{G}(t + 1) = \{\mathcal{G}_i(t + 1)\}_{i=1}^K$, where each $\mathcal{G}_i(t + 1) = (\mu_i, \Sigma_i(t + 1), f_i(t + 1), \alpha_i(t + 1))$ describes a Gaussian distribution sphere. The generation process is defined as:

$$\mathcal{G}_i(t + 1) = G(F(t), p_i, \Theta), \tag{2}$$

where $p_i$ is the RGG point associated with $\mathcal{G}_i$, $\Theta$ represents the parameters in model $G$, and Model is a learnable neural network consisting of the Gaussian embedding layer, spherical graph attention blocks, and the Gaussian decoding layer.

**Gaussian Embedding.** The initial node features are derived from the atmospheric data and positional information, leveraging the raw ERA5 data $F(t)$ at time $t$ sampled at RGG grid points $p_i$. The position $\mu_i \in \mathbb{R}^3$ is derived from the RGG grid's latitude $\phi_i$ and longitude $\lambda_i$, and we apply Fourier positional encoding Tancik et al. (2020) to capture multi-frequency spatial features. The encoded position feature

$p_i \in \mathbb{R}^K$ is defined as:

$$p_i = \left[ \sin\left(\frac{2^k \pi \phi_i}{F_{\max}}\right), \cos\left(\frac{2^k \pi \phi_i}{F_{\max}}\right), \sin\left(\frac{2^k \pi \lambda_i}{F_{\max}}\right), \cos\left(\frac{2^k \pi \lambda_i}{F_{\max}}\right) \right]_{k=0}^{K-1}, \tag{3}$$

where $F$ is the feature dimension (e.g., $2K$ with $K$ frequency bands), and $F_{\max}$ is the maximum frequency scale. This $p_i$ is then passed through a linear layer to obtain $p_i' \in \mathbb{R}^D$: $p_i' = \text{Linear}(p_i)$.

The remaining feature vector $f_i \in \mathbb{R}^{160}$, representing the 160 atmospheric variables sampled from $F(t)$ at $p_i$, is processed through a linear layer to produce $h_f \in \mathbb{R}^D$: $h_f = \text{Linear}(f_i)$. The final $D$-dimensional node feature $h_i^0$ is obtained by adding the position-encoded feature and the atmospheric feature:

$$h_i^0 = p_i' + h_f. \tag{4}$$

**Multi-Scale Attention and Global Teleconnection**  To handle the irregular layout of the RGG grid and model Earth system dynamics, we employ a Graph Attention Transformer (GAT) network Veličković et al. (2018), where nodes correspond to Gaussian points. To capture both local and long-range dependencies in chaotic weather systems, we adopt a multi-scale attention mechanism: GAT blocks model interactions at varying spatial scales, which help capture the global teleconnections.

**Gaussian Decoding.**  The updated features $h_i^L$ after $N$ layers are decoded to predict the parameters of $\mathcal{G}_i(t+1)$. A single linear layer maps $h_i^L$ to a 168-dimensional parameter vector representing the updated Gaussian parameters:

$$\mathbf{p}_i(t+1) = \text{Linear}(h_i^L), \tag{5}$$

where $\mathbf{p}_i(t+1) \in \mathbb{R}^{168}$ encapsulates the quaternion for the rotation matrix $R$, the scaling factors for the diagonal matrix $S$, the 160-dimensional feature vector, and the opacity factor. These parameters are post-processed: the scaling factors are passed through a softplus activation to enforce positivity, the feature vector and opacity factor through a sigmoid activation to constrain them to physically plausible ranges, and the quaternion is normalized to maintain unit length, reconstructing $\Sigma_i(t+1) = RSS^T R^T$. The position $\mu_i(t+1)$ remains fixed (as $\mu_i$ is time-invariant per RGG coordinates), so $\mathcal{G}_i(t+1) = (\mu_i, \Sigma_i(t+1), f_i(t+1), \alpha_i(t+1))$.

## 3.3  Forecast Rendering and Optimization

**Forecast Rendering via Rasterization.**  As shown in Figure 1c), to render the forecasted atmospheric field at time $t+1$, we adopt the reconstruction method in Section 3.1. Specifically, for any query point $p \in S^2$, the atmospheric field $F(p, t+1)$ is reconstructed as a weighted sum of feature vectors modulated by opacity:

$$F(p, t+1) = \sum_{i \in N} f_i(t+1)\alpha_i(t+1)T_i, \tag{6}$$

where $N$ is the set of Gaussians overlapping with $p$, sorted by depth, and $T_i = \prod_{j=1}^{i-1}(1 - \alpha_j(t+1))$ is the transmittance ensuring front-to-back accumulation. To balance reconstruction error and computational cost, we adjust the resolution of the Gaussian splatting by varying the density and coverage of query points $p$. For low-error (high-computation) forecasts, we increase the density of query points to capture fine-grained details, while for higher-error (low-computation) forecasts, we reduce the density. This flexibility leverages the continuous representation of 3DGS and the quasi-uniform RGG grid, enabling GaussianCast to achieve both accurate reconstruction and prediction while saving substantial computational resources.

**End-to-End Optimization.**  Given the differentiable nature of 3DGS rendering, we perform end-to-end supervision by directly comparing the rendered forecast $F(p, t+1)$ with the ground-truth ERA5 data

$\hat{F}(p, t+1)$. The loss function is defined as:

$$\mathcal{L}_{\text{recon}} = \sum_{p \in \text{RGG}} \|F(p, t+1) - \hat{F}(p, t+1)\|_2^2, \tag{7}$$

where $\hat{F}(p, t+1)$ represents the ground-truth atmospheric field from ERA5 at time $t+1$. The model parameters $\Theta$ is optimized to predict Gaussian parameters ($\Sigma_i(t+1)$, $f_i(t+1)$, $\alpha_i(t+1)$). Similarly, generating the Gaussian parameters at time $t$ corresponds to the reconstruction of the atmospheric data.

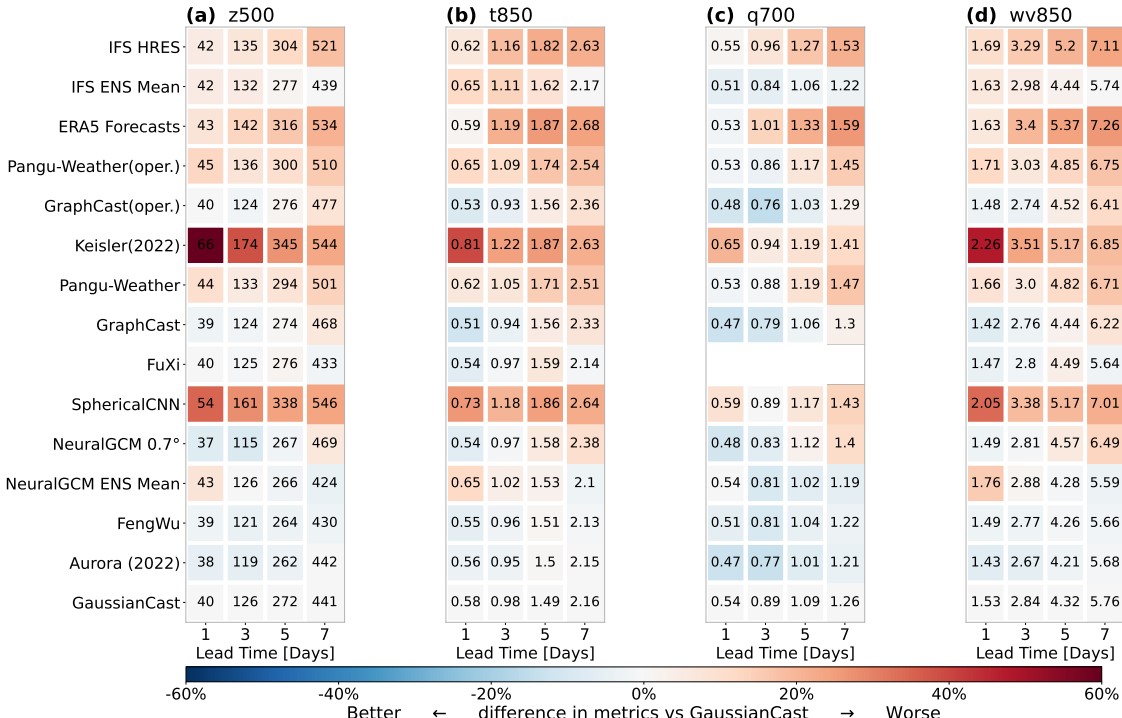

Figure 2: **Scorecard for upper-level variables for the year 2020**. (Quasi-)operational models are evaluated against operational analysis. All other models evaluated against ERA5. All data from WeatherBench, a public Google Cloud bucket: gs://weatherbench2/datasets.

## 4 EXPERIMENTS

### 4.1 IMPLEMENTATION DETAILS

**Dataset.** We conduct experiments on the ERA5 reanalysis dataset Hersbach et al. (2020) from ECMWF, which provides atmospheric variables from 1940 to the present at $0.25°$ resolution ($721 \times 1440$). To reduce computational cost, we train on a 20-year subset (2000–2019) and test on 2020. Our study uses six upper-air variables across 25 levels and 10 surface variables (full list in Appendix).

**Training Details.** The GaussianCast model is trained on 8 NVIDIA A100 80 GB GPUs using a data-parallel configuration. The training process consists of 150k iterations, employing the AdamW optimizer with an

| | Backbone | T2M ($K$) $\downarrow$ | | | U10 ($m/s$) $\downarrow$ | | | V10 ($m/s$) $\downarrow$ | | | MSL ($Pa$) $\downarrow$ | | |
|---|---|---|---|---|---|---|---|---|---|---|---|---|---|
| | | 6h | 72h | 120h | 6h | 72h | 120h | 6h | 72h | 120h | 6h | 72h | 120h |
| ViT* Dosovitskiy (2020) | Transformer | 0.72 | 1.35 | 1.86 | 0.66 | 1.98 | 3.01 | 0.68 | 2.02 | 3.11 | 40.2 | 208.5 | 393.9 |
| IFS Lang et al. (2024a) | Physics | 1.09 | 1.38 | 1.74 | 0.96 | 1.87 | 2.78 | 0.99 | 1.93 | 2.87 | - | - | - |
| Pangu-Weather Bi et al. (2022) | Transformer | 0.82 | 1.09 | 1.53 | 0.77 | 1.63 | 2.54 | 0.79 | 1.68 | 2.65 | - | - | - |
| FourCastNet Pathak et al. (2022) | AFNO | 0.82 | 1.02 | 1.77 | 0.82 | 2.08 | 3.34 | 0.84 | 2.11 | 3.41 | - | - | - |
| ClimaX Nguyen et al. (2023) | Transformer | 1.11 | 1.47 | 1.83 | 1.04 | 2.02 | 2.79 | - | - | - | - | - | - |
| Graphcast Lam et al. (2023) | GNN | **0.51** | **0.94** | 1.37 | **0.38** | 1.51 | 2.37 | - | - | - | 23.4 | 135.2 | 278.2 |
| Fengwu Chen et al. (2023a) | Transformer | 0.58 | 1.03 | 1.41 | 0.42 | 1.53 | 2.32 | - | - | - | **23.2** | 137.1 | 276.9 |
| FuXi Chen et al. (2023c) | Transformer | 0.55 | 0.99 | 1.41 | 0.42 | 1.50 | 2.36 | **0.43** | 1.54 | 2.44 | 27.2 | 136.7 | 282.9 |
| VA-MoE Chen et al. (2024) | Transformer | 0.57 | 1.03 | 1.42 | 0.43 | **1.41** | **2.25** | 0.44 | **1.46** | **2.34** | 27.5 | 131.1 | 275.9 |
| Aurora Bodnar et al. (2025) | Transformer | 0.53 | 0.96 | **1.32** | 0.69 | 1.55 | 2.29 | 0.69 | 1.60 | 2.38 | 32.6 | **130.5** | **268.0** |
| GaussianCast | GAT+3DGS | 0.56 | 0.98 | 1.45 | 0.42 | 1.46 | 2.28 | 0.44 | 1.50 | 2.46 | 25.4 | 138.4 | 279.8 |

Table 1: The RMSE of 5 Surface-level variables, i.e., T2M, U10, V10, MSL. The best results are marked in **bold**. **All experiments are in 0.25°with** $721 \times 1440$ **resolutions.**

initial learning rate of $1 \times 10^{-4}$. The learning rate is decayed using a cosine schedule to $1 \times 10^{-6}$ over the first 100k iterations and remains constant at $1 \times 10^{-6}$ for the remaining 50k iterations. The implementation is based on DeepSpeed's training framework, taking 10 days to complete the training process.

**Evaluation Setup.** The model's performance is evaluated on nine key atmospheric variables: 2-meter temperature (T2m), 10-meter zonal wind (U10), 10-meter meridional wind (V10), mean sea level pressure (MSL), geopotential at 500 hPa (Z500), temperature at 850 hPa (T850), specific humidity at 700 hPa (Q700), wind speed ($\sqrt{U850^2 + V850^2}$) at 850 hPa (wind850). Forecast accuracy is measured using the latitude-weighted root-mean-square error (RMSE) Pathak et al. (2022); Han et al. (2024), which accounts for the varying area of grid cells with latitude to provide a more representative error metric across the globe (See Appendix for definition). The evaluation spans lead times ranging from 1 to 7 days. GaussianCast is pretrained with a 6-hour interval, and to achieve long-term predictions, autoregressive prediction is employed for forecasts from 1 to 7 days.

### 4.2 COMPARISON OF MEDIUM-RANGE WEATHER FORECASTS

**Upper-level Evaluation.** The upper-level evaluation was conducted on four variables—Z500, T850, Q700, wind850—using the ERA5 dataset in 2020 year, with latitude-weighted RMSE scores for lead times of 1, 3, 5, and 7 days, as shown in the scoreboard in Figure 2. GaussianCast achieves RMSE values of 40 m$^2$/s$^2$ for Z500, 0.58 K for T850, 0.54 g/kg for Q700, and 1.53 m/s for wind850 at 1-day lead time, outperforming Pangu-Weather Bi et al. (2022) , SphericalCNN Esteves et al. (2023), and Keisler (2022) Keisler (2022). At 7 days, GaussianCast's scores (441 m$^2$/s$^2$ for Z500) are competitive with the best meteorological models like FengWu Chen et al. (2023a) (430 m$^2$/s$^2$) and FuXi Chen et al. (2023c) (433 m$^2$/s$^2$), and significantly outperform Pangu Bi et al. (2022) (501 m$^2$/s$^2$) GraphCast Lam et al. (2023) (468 m$^2$/s$^2$). These results indicate that the proposed Gaussian distribution-based forecasting approach is not only feasible but also exhibits strong potential to surpass existing large-scale meteorological models.

**Surface Evaluation.** We evaluate 4 surface variables—2-meter temperature (T2m), 10-meter zonal wind (U10), 10-meter meridional wind (V10), and mean sea-level pressure (MSL), with RMSE computed for lead times of 6 hours, 72 hours, and 120 hours, as shown in Table 1. Unlike other AI weather forecasting models, GaussianCast is a rasterization-based method, achieving competitive RMSEs from 6 hours to longer lead times. Compared to the best models, such as GraphCast Lam et al. (2023), FengWu Chen et al. (2023a), and VA-MoE

Chen et al. (2024), GaussianCast performs comparably, demonstrating that its Gaussian distribution-based approach is viable and holds potential to rival or exceed current state-of-the-art models.

### 4.3 COMPARISON OF MODEL COMPLEXITY

GaussianCast, with 40M parameters and 1843 G FLOPs, is significantly smaller and less computationally intensive than baselines like Pangu-Weather Bi et al. (2022), FuXi Chen et al. (2023c), and FengWu Chen et al. (2023a), as shown in Table 2. it achieves comparable performance, with a T2M RMSE of 0.56 K at 6-hour lead time versus GraphCast's 0.51 K, highlighting its efficiency in delivering high accuracy with minimal resources.

| Model | Parameter Size | FLOPs |
|---|---|---|
| Pangu-Weather | 740M | 4950 G |
| GraphCast | 55M | 13564 G |
| FuXi | 4B | 19892 G |
| FengWu | 751M | 8000 G |
| GaussianCast | 101M | 1843 G |

Table 2: Model parameters and FLOPs.

### 4.4 ABLATION

**The number of Reduced Gaussian Grids.** To assess the impact of the number of Reduced Gaussian Grid (RGG) points on 3D Gaussian Splatting (3DGS) atmospheric data representation, we conducted an ablation study with 49493, 57617, 69185, 86717, and 116449 points, as shown in Figure 3, which plots the latitude-weighted RMSE for Z500, T850, V850, and MSL over 4000 iterations. Results indicate that increasing point numbers reduces RMSE (e.g., Z500 from 16 to 4 $m^2/s^2$, T850 from 0.30 to 0.15 K), improving representation fidelity, but also raises computational complexity; thus, we selected 69185 points as the optimal trade-off, achieving Z500 RMSE < 6 $m^2/s^2$ and T850 RMSE < 0.2 K, with additional variable performances detailed in the Appendix.

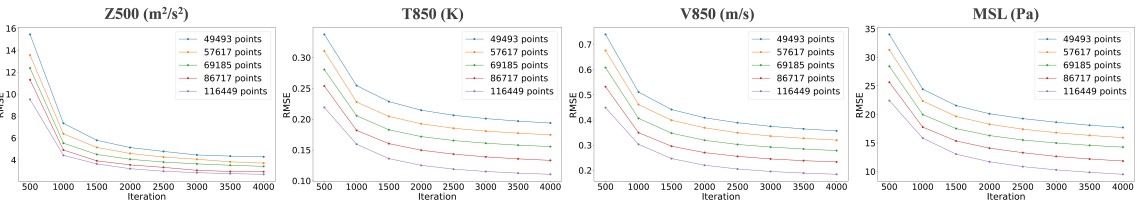

Figure 3: Reconstruction performance of 3DGS with varying RGG point numbers.

### 4.5 EXTRAPOLATION OF MULTI-SCALE FORECASTS

The distribution-based modeling is capable of multi-scale forecasts, we here show GaussianCast's prediction rendering at different spatial resolutions, as shown in Figure 4. Despite being trained on ground-truth ERA5 data at 0.25° resolution, our model successfully generates high-resolution forecasts at 0.1°, capturing finer details and improving visual quality. For instance, at 0.1° resolution, GaussianCast effectively eliminates jagged artifacts (e.g., aliasing effects) commonly observed in 1° and 0.25° predictions, particularly in regions with sharp gradients such as frontal boundaries and storm systems. These results validate GaussianCast's unique advantage in arbitrary-scale rendering, aligning with our contribution of supporting multi-resolution predictions without retraining, thus addressing the limitations of fixed-resolution AI-based weather models. See Appendix for more visualizations.

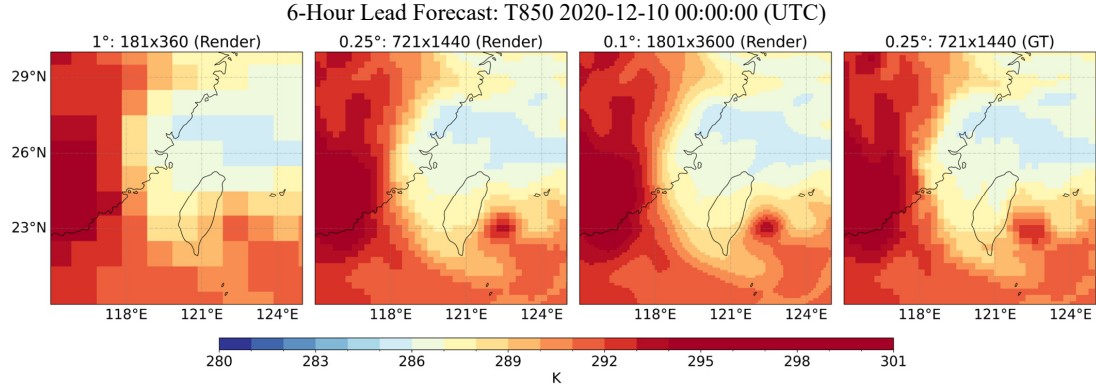

Figure 4: Multi-scale forecasting results of GaussianCast at 1° (181×360), 0.25° (721×1440), and 0.1° (1801×3600) resolutions. Despite training on 0.25° ground-truth ERA5 data, the model achieves high-fidelity predictions at 0.1°, with improved details and reduced aliasing effects.

## 4.6 TROPICAL CYCLONE TRACKING

To evaluate GaussianCast's performance in typhoon forecasting, we predict the first typhoon of 2025, Typhoon Wutip. Figure 5 presents the RMSE of the predicted typhoon track in comparison with other methods. As shown in the figure, our method significantly reduces the RMSE compared with traditional numerical weather prediction models. Specifically, at 120 hours ahead, GaussianCast achieves an 85% lower RMSE than ECMWF and NCEP. Furthermore, GaussianCast demonstrates performance comparable to AI-based methods, including Pangu-Weather Bi et al. (2022), FuXi Chen et al. (2023c), and GraphCast Lam et al. (2023), effectively combining the strengths of data-driven modeling with physical consistency.

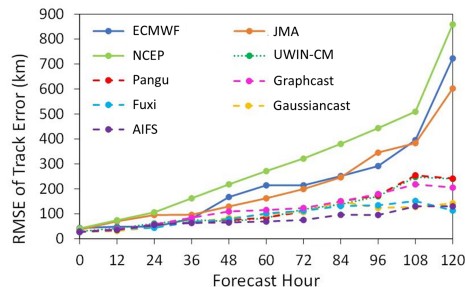

Figure 5: Tropical cyclone track errors against IBTrACS observations.

## 5 CONCLUSION

GaussianCast advances numerical weather prediction (NWP) by integrating 3D Gaussian Splatting (3DGS) with a Reduced Gaussian Grid (RGG), generating a compact and continuous representation of atmospheric fields that mitigates data redundancy and achieves up to 14× compression. Conditioned on the current state, multi-scale Graph Attention Transformers generate Gaussian parameters for reconstruction and medium-range forecasting, enabling accurate weather reconstruction and skillful forecasts at substantially lower computational cost. Experiments on the ERA5 dataset demonstrate that GaussianCast performs competitively against state-of-the-art models such as GraphCast and FengWu, while surpassing larger models like Pangu-Weather and FuXi in efficiency, producing 10-day forecasts in just 20 seconds on an A100 GPU. These findings underscore GaussianCast's potential to bridge AI-driven forecasting and traditional NWP, offering a efficient and interpretable solution for next-generation weather prediction.

## ETHICS STATEMENT

We affirm that this research complies with the ICLR Code of Ethics. Our study does not involve human subjects, personal or sensitive data, or animal experiments. The methods and datasets used are either publicly available or synthetic, and do not raise concerns regarding privacy, security, discrimination, bias, or fairness. There are no conflicts of interest, sponsorship influences, or legal compliance issues associated with this work. All results and findings are reported transparently and honestly. We have thoroughly reviewed the ICLR Code of Ethics and confirm that our research adheres to its guidelines.

## REPRODUCIBILITY STATEMENT

We are committed to ensuring the reproducibility of our results. All implementation details, including model architectures, training procedures, and hyperparameters, are described in detail in the main text and Appendix. The datasets used are publicly available, and all data preprocessing steps are clearly explained in the Appendix materials. We provide comprehensive experimental settings and evaluation protocols to facilitate replication. Additionally, we supply an anonymous link to the source code and scripts necessary to reproduce our experiments in the Appendix materials. We encourage readers to refer to the main paper and Appendix files for all information required to fully reproduce our results.

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

## A  USAGE OF LLMs

We used large language models (LLMs) solely as a general-purpose assistive tool for language polishing and improving readability. The LLMs did not contribute to research ideation, experimental design, or the generation of scientific content. All scientific claims, analyses, and conclusions in this paper are entirely the responsibility of the authors.

## B  EXTRA EXPERIMENTS ANALYSIS

### B.1  COMPLETE LIST OF VARIABLES

As shown in Table 3 and Table 4, this study selects six upper-air variables (Z, Q, U, V, T, W) across 25 pressure levels (1, 2, 3, 5, 7, 10, 20, 30, 50, 70, 100, 150, 200, 250, 300, 400, 500, 600, 700, 800, 850, 900, 925, 950, 1000 hPa), representing geopotential height, specific humidity, zonal wind, meridional wind, temperature, and vertical velocity, respectively. Additionally, ten surface variables are analyzed: 10m wind components (U10, V10), 100m wind components (U100, V100), 2m temperature (T2M), 2m dewpoint temperature (D2M), total cloud cover (TCC), mean sea-level pressure (MSL), 6-hour accumulated precipitation (TP6H), and 6-hour accumulated surface solar radiation (SSR6H).

| Symbol | Full Name | Symbol | Full Name |
|---|---|---|---|
| Z | Geopotential Height | V | Meridional Wind |
| Q | Specific Humidity | T | Temperature |
| U | Zonal Wind | W | Vertical Velocity |

Table 3: A summary of Upper-level atmospheric variables.

| Symbol | Full Name | Symbol | Full Name |
|---|---|---|---|
| V10 | 10m Meridional Wind | TCC | Total Cloud Cover |
| U10 | 10m Zonal Wind | D2M | 2m Dewpoint |
| V100 | 100m Meridional Wind | MSL | Mean Sea Level Pressure |
| U100 | 100m Zonal Wind | TP6H | 6h Total Precipitation |
| T2M | 2m Temperature | SSR6H | 6h Surface Solar Radiation |

Table 4: A summary of Surface-level atmospheric variables.

### B.2  EVALUATIONS METRICS

The Latitude-Weighted Root-Mean-Square Error (WRMSE) addresses the distortion of grid cell areas in latitude-longitude coordinate systems by assigning cosine-latitude weights. For a global field with $N$ grid points, WRMSE is computed as:

$$\text{WRMSE} = \sqrt{\frac{1}{\sum_{i=1}^{N} w_i} \sum_{i=1}^{N} w_i \cdot (y_i - \hat{y}_i)^2} \tag{8}$$

where $y_i$ and $\hat{y}_i$ are the observed and predicted values at grid point $i$; $w_i = \cos(\phi_i)$ is the weight for grid point $i$; $\phi_i$ is the latitude (in radians) of grid point $i$'s center.

This weighting scheme ensures balanced error contributions across latitudes, as unweighted RMSE would disproportionately emphasize high-latitude grid cells where longitudinal lines converge. The $\cos(\phi)$ weighting exactly compensates for the reducing actual area of grid cells in equal-angle latitude-longitude grids.

## C  MULTI-SCALE GRAPH ATTENTION TRANSFORMER ARCHITECTURE

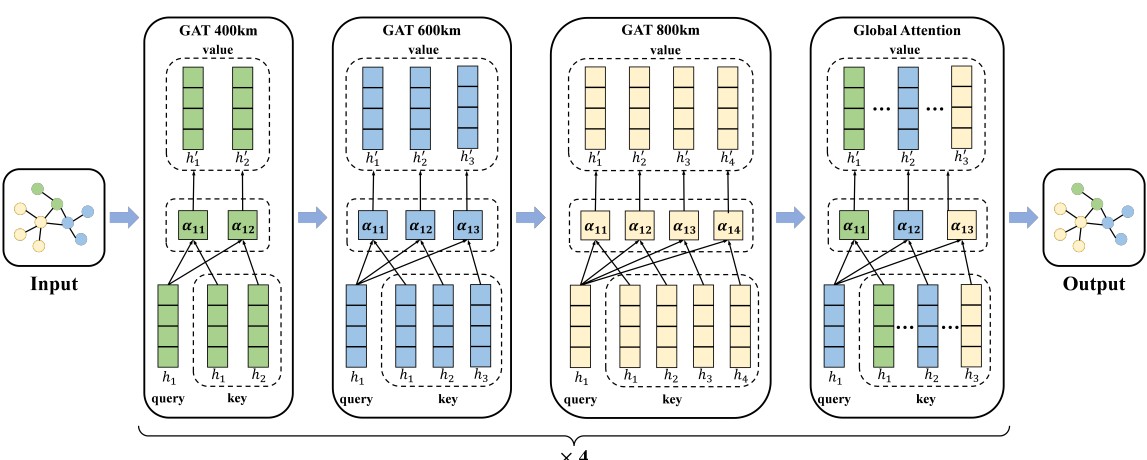

Figure 6: Architecture of the proposed multi-scale GAT variants. From left to right, the attention receptive fields cover spatial ranges of 400 km, 600 km, 800 km, and a global scale.

As shown in Figure 6, our 16-layer architecture consists of four repeated blocks, each comprising four Graph Attention Transformer (GAT) layers with progressively increasing spatial receptive fields. Each block includes:

(1) GAT 400 km: Captures local-scale interactions within a 400 km radius, focusing on fine-grained spatial dependencies and rapid variations.

(2) GAT 600 km: Expands the receptive field to 600 km to incorporate short-range regional interactions, enabling the aggregation of more spatially distributed patterns.

(3) GAT 800 km: Extends the spatial scope to 800 km, allowing the model to capture broader regional dependencies beyond the immediate vicinity.

(4) Global Attention: Employs fully connected attention across all locations, enabling the modeling of long-range dependencies and global-scale interactions.

This hierarchical structure is repeated four times, enabling iterative refinement of spatial representations across multiple scales. Residual connections and adaptive normalization are applied between layers to ensure stable optimization and efficient information flow. The expanding-then-resetting design reflects the inherently multi-scale nature of spatiotemporal processes, where both local and distant interactions contribute to system evolution.

We adopt multi-scale GAT blocks as the basic Transformer module in our networks. Here, we provide detailed architectural specifications of the multi-scale GAT configurations at different spatial scales, as summarized in Table 5.

| Stage | GAT Module | Output Size |
|---|---|---|
| Input Data | Latitude and longitude$[x, y]$ Features$[f_1, ..., f_{160}]$ | $B \times N \times 162$ |
| Fourier Encode | Fourier Positional Encoding (64 freq) | $B \times N \times 256$ |
| Feature Fusion | Fea. Linear(160) + Pos. Linear(256) $\rightarrow$ 512-dim | $B \times N \times 512$ |
| Encoder | **Repeat 4 times:**
SpatialGATLayer(edge_index$_i$)
SpatialGATLayer(edge_index$_i$)
SpatialGATLayer(edge_index$_i$)
GlobalAttentionLayer
Mask index: $i = $ (layer index $\mathrm{mod}$ 4)
Edge_index$_i$ corresponds to distance thresholds of 400km ($i = 0$),
600km ($i = 1$), and 800km ($i = 2$) (used only if $i <$ num_masks) | $B \times N \times 512$ |
| Post Norm | LayerNorm(512) | $B \times N \times 512$ |
| Output Proj. | Linear(512, no bias) $\rightarrow$ 168 | $B \times N \times 168$ |

Table 5: Architecture overview of Graph Attention Transformer. Encoder block repeats every 4 layers, mixing multi-scale SpatialGAT and Global Attention layers.

## D  BROADER IMPACTS

Our GaussianCast introduces an efficient and highly practical atmospheric forecasting approach, significantly improving access to advanced weather modeling for various forecasting agencies. Its compact and computationally lightweight design enables high-fidelity forecasts on modest hardware, democratizing state-of-the-art weather prediction for resource-constrained regions, national weather services, and local disaster response agencies.

Moreover, GaussianCast's interpretable 3D Gaussian representation explicitly encodes physical attributes and uncertainties, enhancing forecast transparency and explainability. In particular, this explainability facilitates improved interpretation of extreme weather events, enabling clearer identification of underlying physical processes, better-informed risk assessment, and more effective communication of forecasts to stakeholders and the broader public.

## E  LIMITATIONS.

GaussianCast faces several limitations that warrant further investigation. First, the computational efficiency advantage diminishes when scaling to global teleconnection modeling with Transformer blocks, as the self-attention mechanism introduces quadratic complexity with respect to the number of nodes. Additionally, the current framework lacks explicit uncertainty quantification in the generative process, which could enhance interpretability and decision-making in operational settings. Future work should explore optimized attention mechanisms for global interactions, and probabilistic modeling to provide confidence intervals for forecasts.

# F    MORE VISUALIZATION RESULTS

## F.1    ABLATION VISUALIZATION

To further analyze the representational capacity of 3D Gaussian Splatting (3DGS) across different atmospheric variables and vertical layers, we extend the ablation study in the main paper by examining how reconstruction error evolves during training. Specifically, Figure 7 illustrates RMSE trajectories over 0–4000 training iterations for four representative variables—specific humidity, temperature, zonal wind, and geopotential height—evaluated across 25 standard pressure levels (1000 hPa to 1 hPa). This analysis aims to reveal how the fidelity of 3DGS-based representation varies with both variable type and pressure levels.

The overall RMSEs across atmospheric pressure levels remain consistently low, underscoring the robustness of GaussianCast's forecasting capability throughout the vertical profile. For example, temperature RMSE range from approximately 0.045 K at upper levels (e.g., 50 hPa) to around 0.275 K near the surface (e.g., 1000 hPa), while wind speed RMSE stay within 0.08–0.30 m/s across most levels. Although minor differences in error magnitudes are observed between different height levels, these are well within acceptable bounds and primarily reflect the natural variation in the scale of each variable. Specifically, temperature variables at lower pressure levels (e.g., 850–1000 hPa) exhibit larger RMSE variability, ranging from 0.15 to 0.28 K, whereas those at higher levels (e.g., 1–7 hPa) show much smaller variation, typically between 0.04 and 0.05 K. Therefore, slightly higher RMSE at lower levels are expected and do not indicate degradation in model performance. Instead, they reflect the inherent physical complexity and broader value ranges near the surface. These findings confirm that GaussianCast maintains stable accuracy across the atmospheric column and handles the multiscale nature of meteorological variables effectively.

## F.2    FORECASTS VISUALIZATION

Figure 8 visualizes the 6-hour global forecasts for upper-air variables, including geopotential height (Z500), temperature (T850), specific humidity (Q700), and vertical wind velocity (V850), with initial conditions at 2020-07-01 00:00 UTC. Figure 9 presents additional upper-air variables: zonal wind at 850 hPa (U850) and temperature at 500 hPa (T500), along with surface variables, namely total cloud cover (TCC) and 6-hour accumulated total precipitation (TP6H). Figure 10 shows the 6-hour global forecasts for surface variables, including 2-meter temperature (T2M), 10-meter wind components (U10, V10), and mean sea level pressure (MSL), with the same initial conditions. These visualizations collectively demonstrate the model's capability to produce accurate forecasts across both upper-air and surface variables at global scale.

In Figure 11, we further illustrate the capability of GaussianCast by providing comprehensive visualizations at varying spatial point densities, including 49,493, 57,617, 69,185, 86,717, and 116,449 points. As the number of spatial points increases, GaussianCast progressively refines its predictions, enabling the capture of increasingly fine-grained atmospheric structures while maintaining physical consistency across scales. These results demonstrate that the model not only remains stable but also improves resolution fidelity as spatial density increases. This highlights GaussianCast's strong scalability and flexibility.

Additionally, we provide more visualization results to showcase GaussianCast's generalization capability across diverse meteorological variables. These include humidity at the 1000 hPa level, wind speed, as well as the Z (geopotential height) and W (vertical velocity) variables. For surface-level analysis, we also include visualizations of total precipitation over 6 hours (TP6H), mean sea level pressure (MSL), and total cloud cover (TCC). These extended results further validate the model's robustness and effectiveness in handling a wide range of atmospheric variables under different resolutions.

Specifically, Figure 11 presents the forecasting results for the temperature variable at the 850 hPa pressure level, while Figure 12 shows the specific humidity (Q) at the 1000 hPa level. Figure 13 illustrates the 1000 hPa wind speed field, and Figure 14 depicts the geopotential height (Z) at the same level. Vertical wind

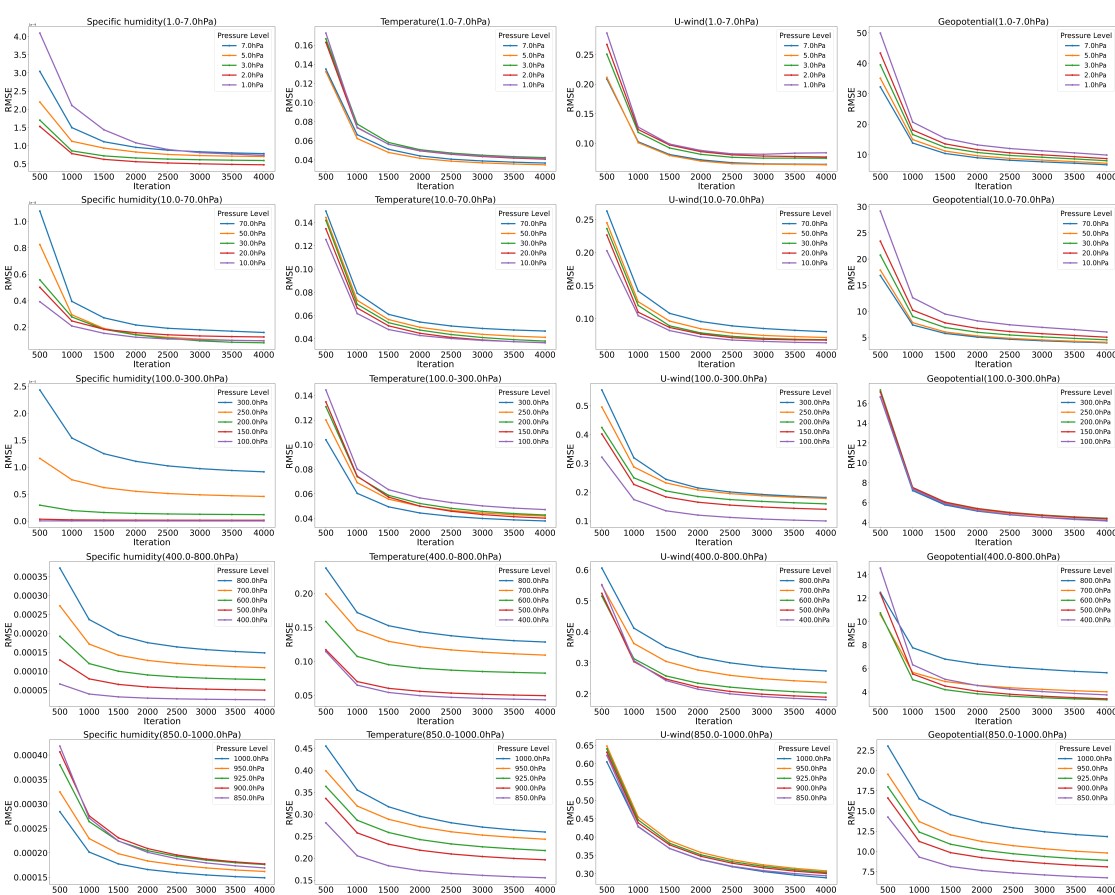

Figure 7: Reconstruction performance of 3DGS with varying pressure levels.

velocity (W) at 1000 hPa is shown in Figure 15. For surface variables, we visualize total precipitation over 6 hours (TP6H) in Figure 16, mean sea level pressure (MSL) in Figure 17, and total cloud cover (TCC) in Figure 18. These visualizations collectively demonstrate the model's ability to handle diverse physical variables across vertical levels and surface layers, reinforcing its applicability to real-world, high-resolution operational forecasting scenarios.

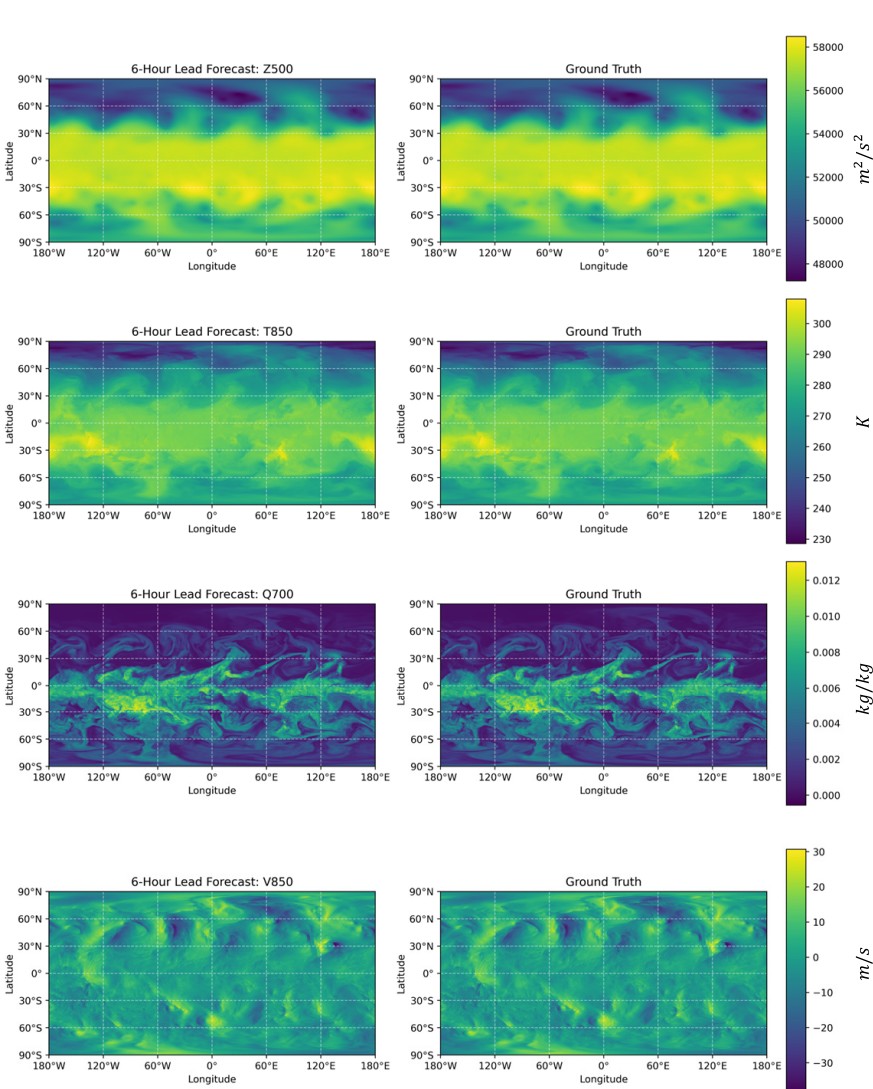

Figure 8: Visualization of the model's 6-hour global forecast of upper-air variables, including geopotential height at 500 hPa (Z500), temperature at 850 hPa (T850), specific humidity at 700 hPa (Q700), and wind velocity at 850 hPa (V850), initialized at 00:00 UTC on July 1st, 2020, with comparison to the ERA5 ground truth.

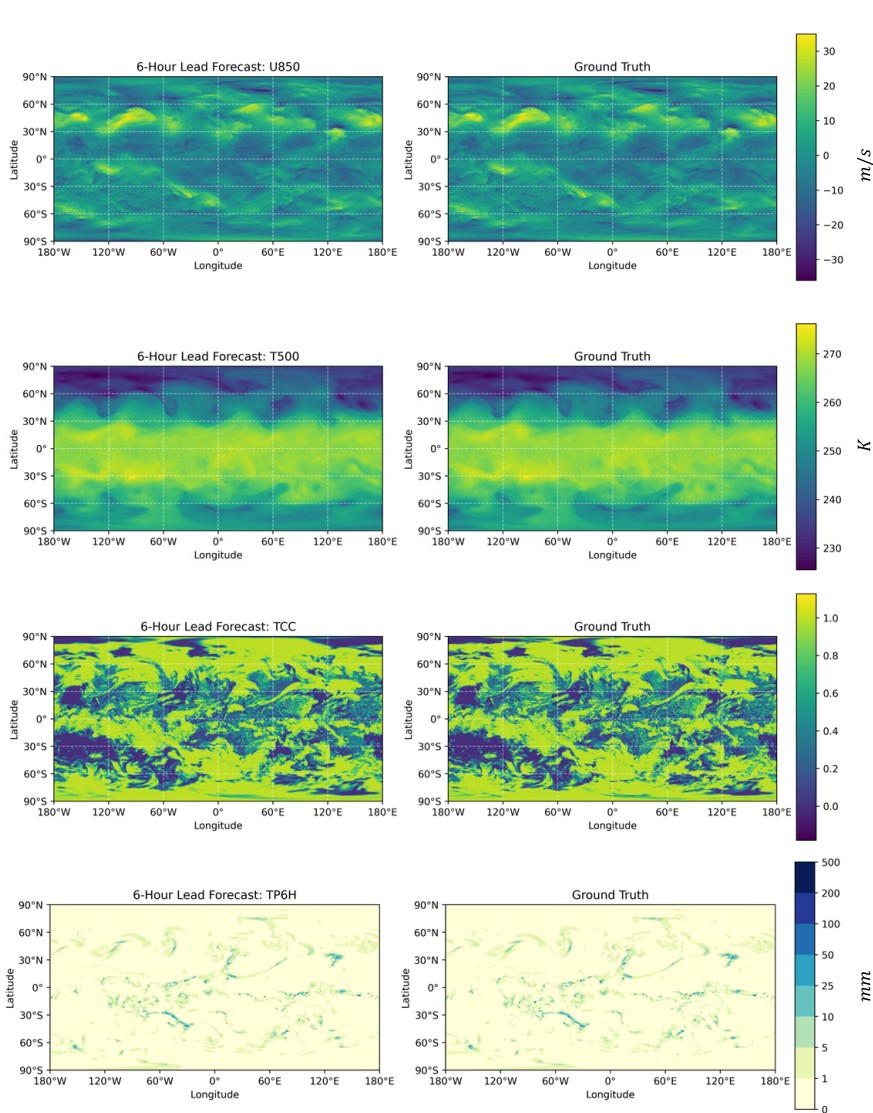

Figure 9: Visualization of the model's 6-hour global forecast of upper-air and surface variables, including wind velocity at 850 hPa (U850), temperature at 500 hPa (T500), total cloud cover (TCC) and total precipitation over 6 hours (TP6H), initialized at 00:00 UTC on July 1st, 2020, with comparison to the ERA5 ground truth.

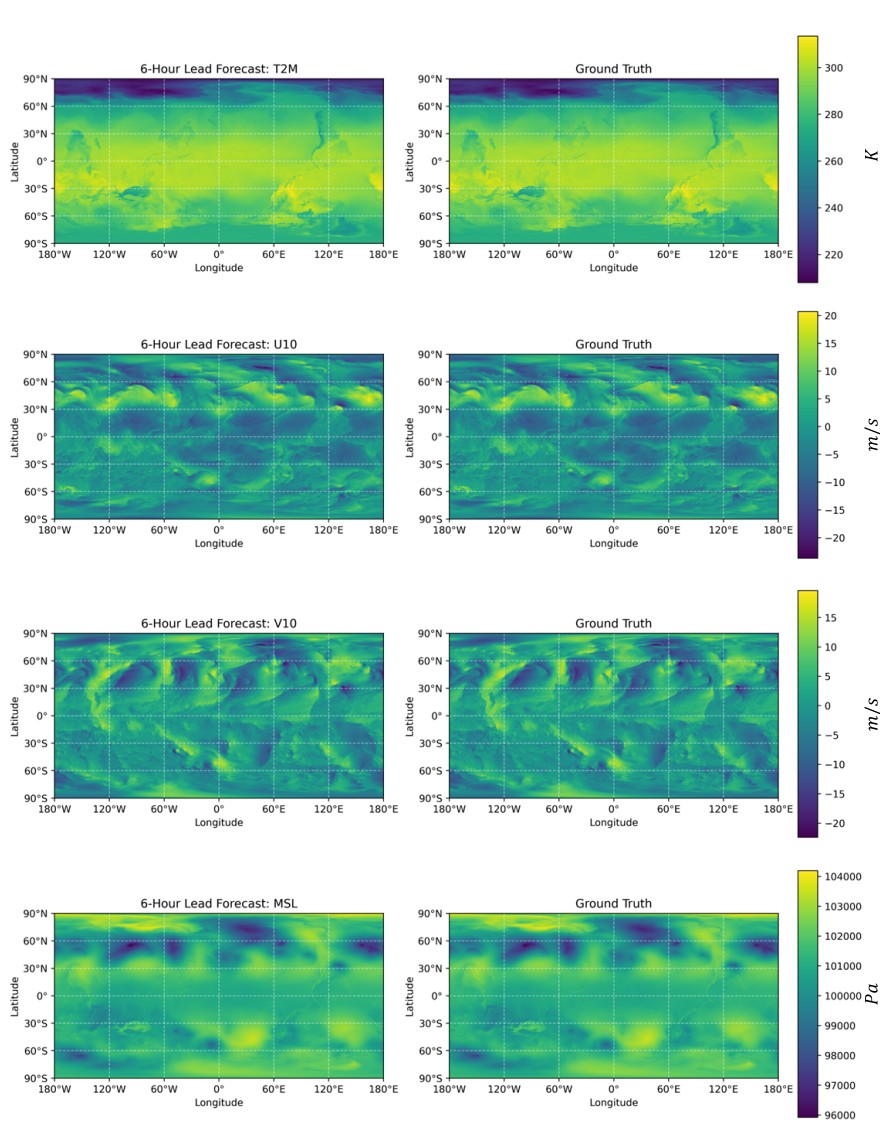

Figure 10: Visualization of the model's 6-hour global forecast of surface variables, including 2-meter temperature (T2M), 10-meter wind components (U10 and V10), and mean sea level pressure (MSL), initialized at 00:00 UTC on July 1st, 2020, with comparison to the ERA5 ground truth.

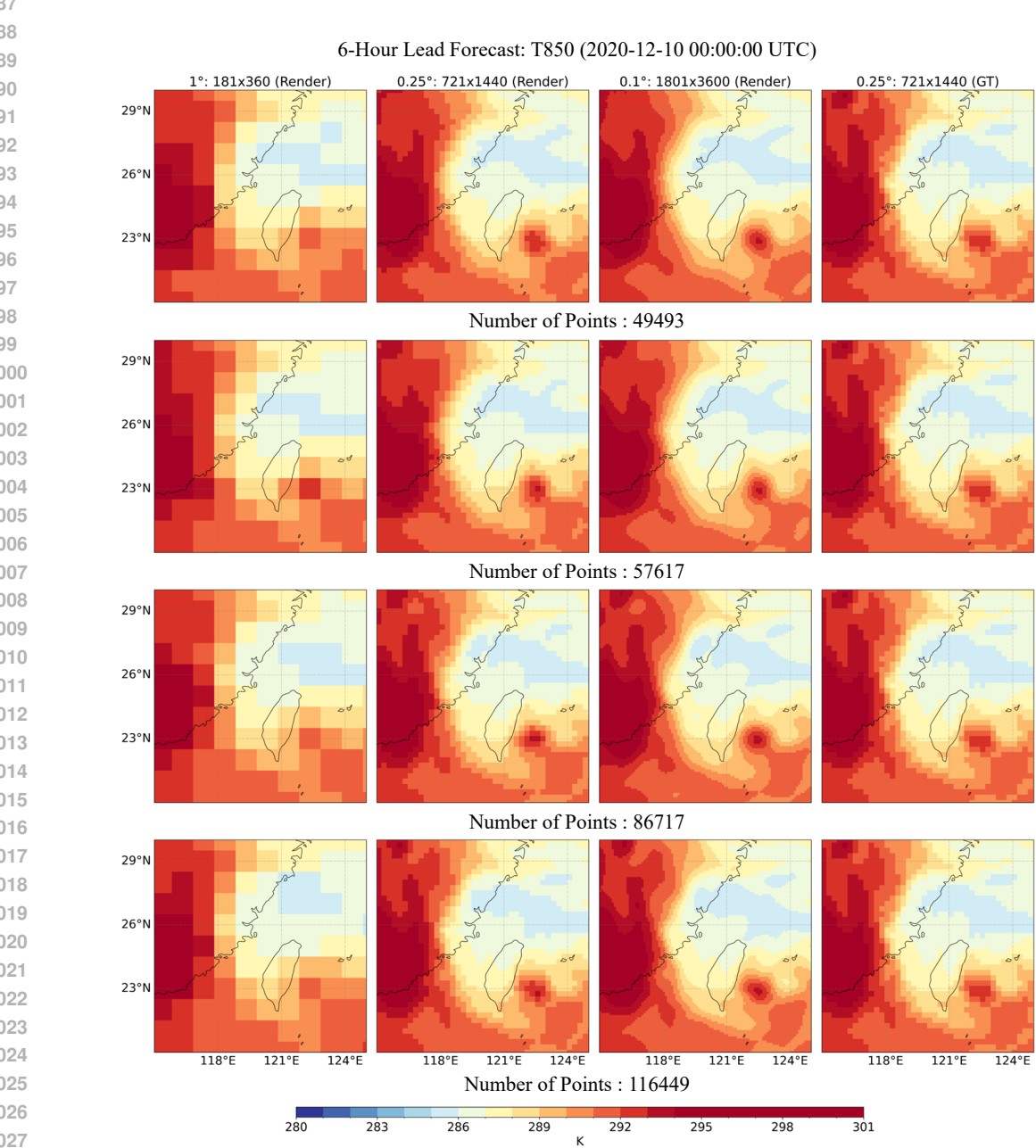

Figure 11: Multi-scale forecasting results of GaussianCast for temperature (T) at the 850 hPa level at 1° (181×360), 0.25° (721×1440), and 0.1° (1801×3600) resolutions with varying numbers of Gaussian points. While trained on 0.25° ERA5 ground truth data, the model achieves high-fidelity predictions at 0.1° resolution, with progressively refined thermal structures as the point count increases.

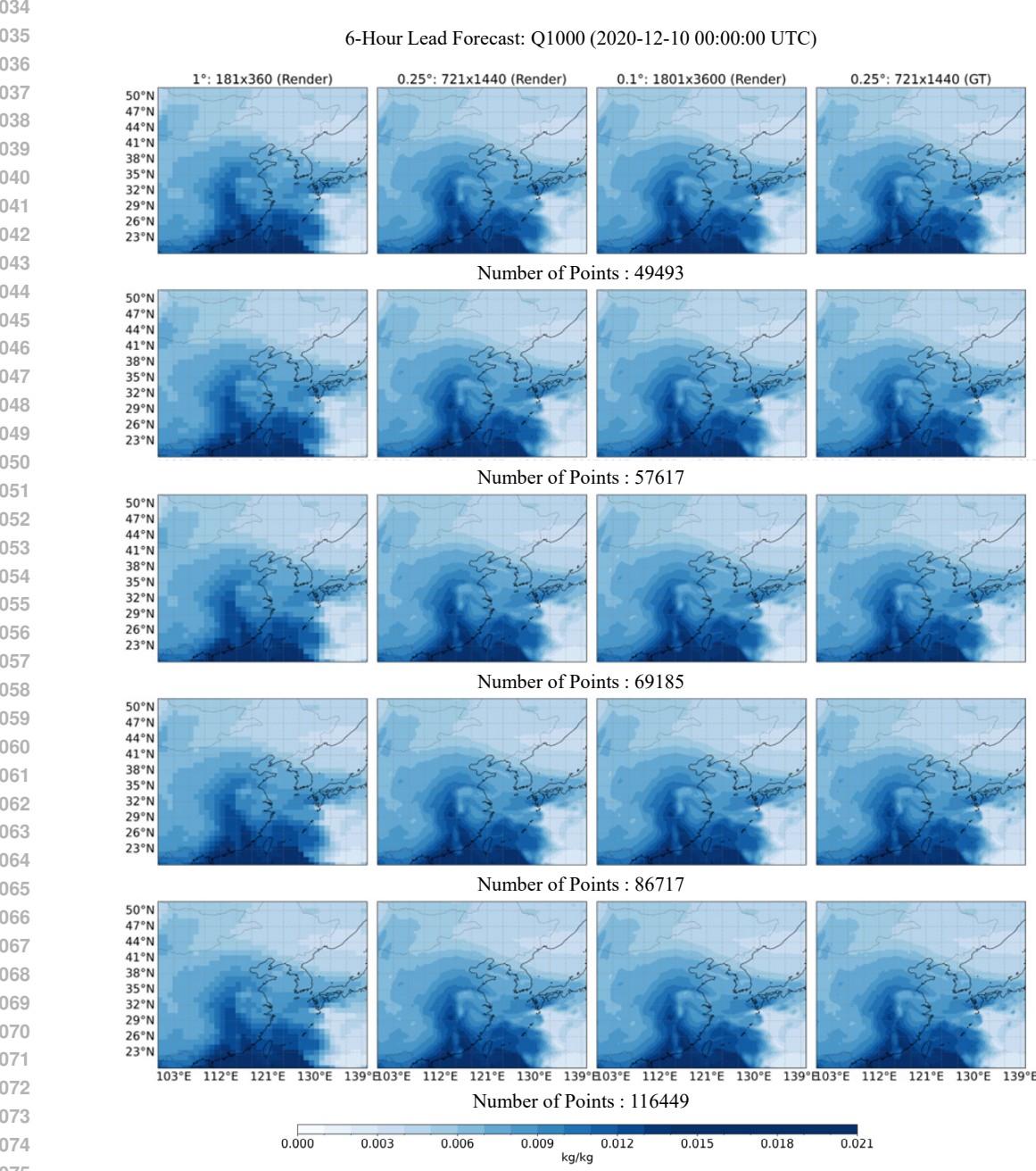

Figure 12: Multi-scale forecasting results of GaussianCast for specific humidity (Q) at the 1000 hPa level at 1° (181×360), 0.25° (721×1440), and 0.1° (1801×3600) resolutions with varying numbers of Gaussian points. While trained on 0.25° ERA5 ground truth data, the model captures sharper moisture gradients and convective features at 0.1° resolution as the point count increases.

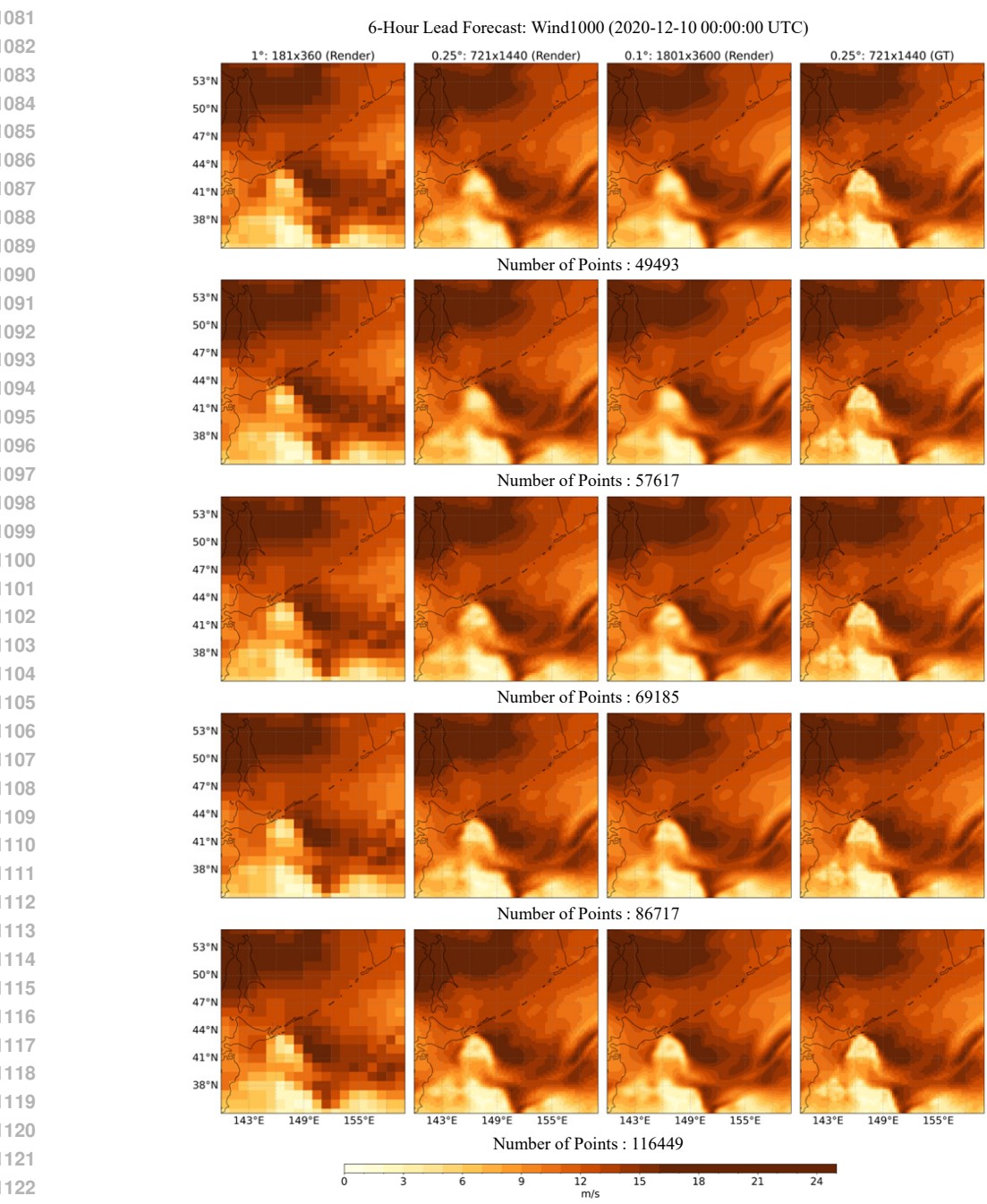

Figure 13: Multi-scale forecasting results of GaussianCast for wind speed at the 1000 hPa level at 1° (181×360), 0.25° (721×1440), and 0.1° (1801×3600) resolutions with varying numbers of Gaussian points. While trained on 0.25° ERA5 ground truth data, the model effectively recovers detailed wind patterns and coherent flow structures at 0.1° resolution with increasing point density.

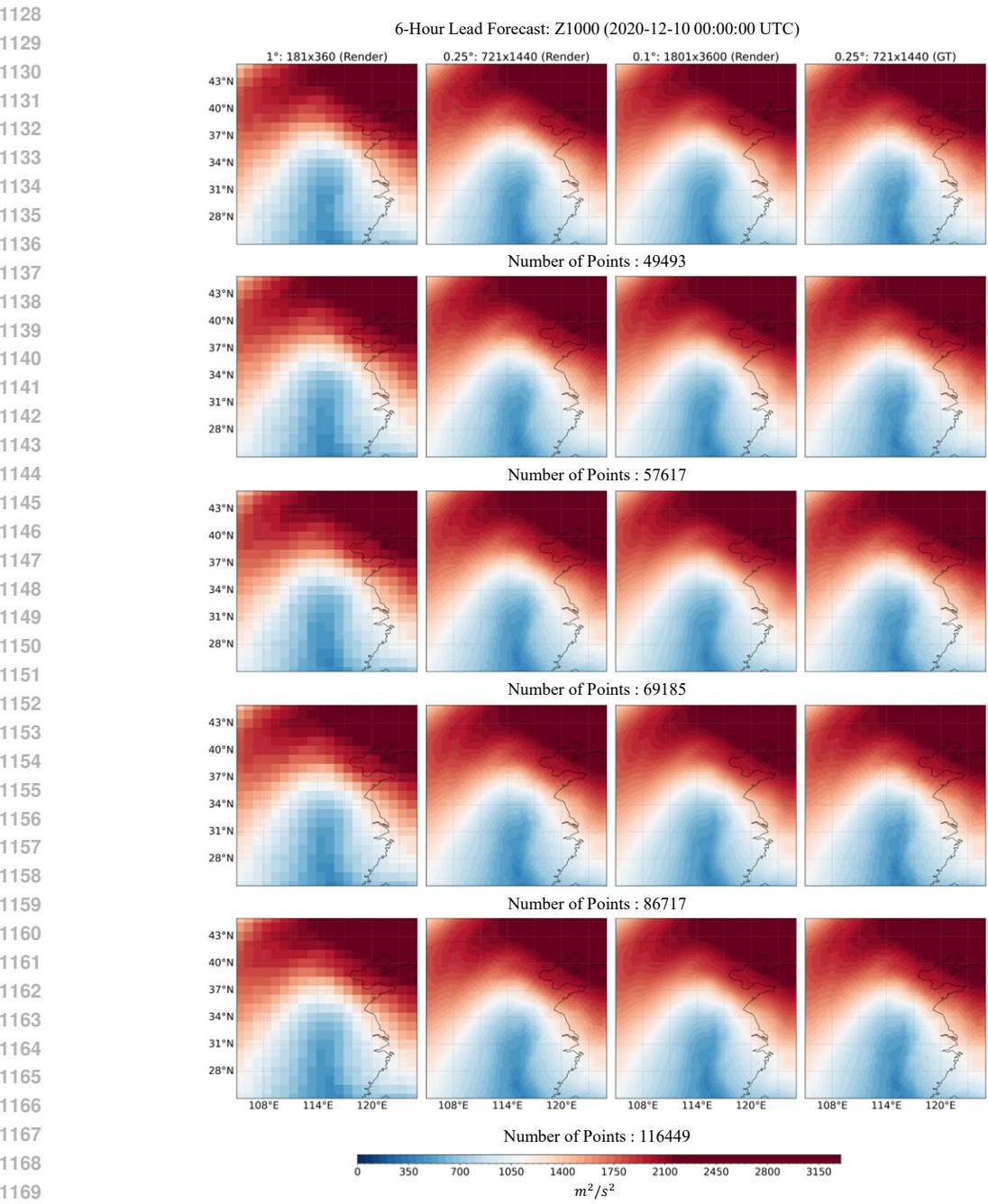

Figure 14: Multi-scale forecasting results of GaussianCast for geopotential height (Z) at the 1000 hPa level at 1° (181×360), 0.25° (721×1440), and 0.1° (1801×3600) resolutions with varying numbers of Gaussian points. While trained on 0.25° ERA5 ground truth data, the model reproduces smooth and fine-scale pressure patterns at 0.1° resolution with more Gaussian points.

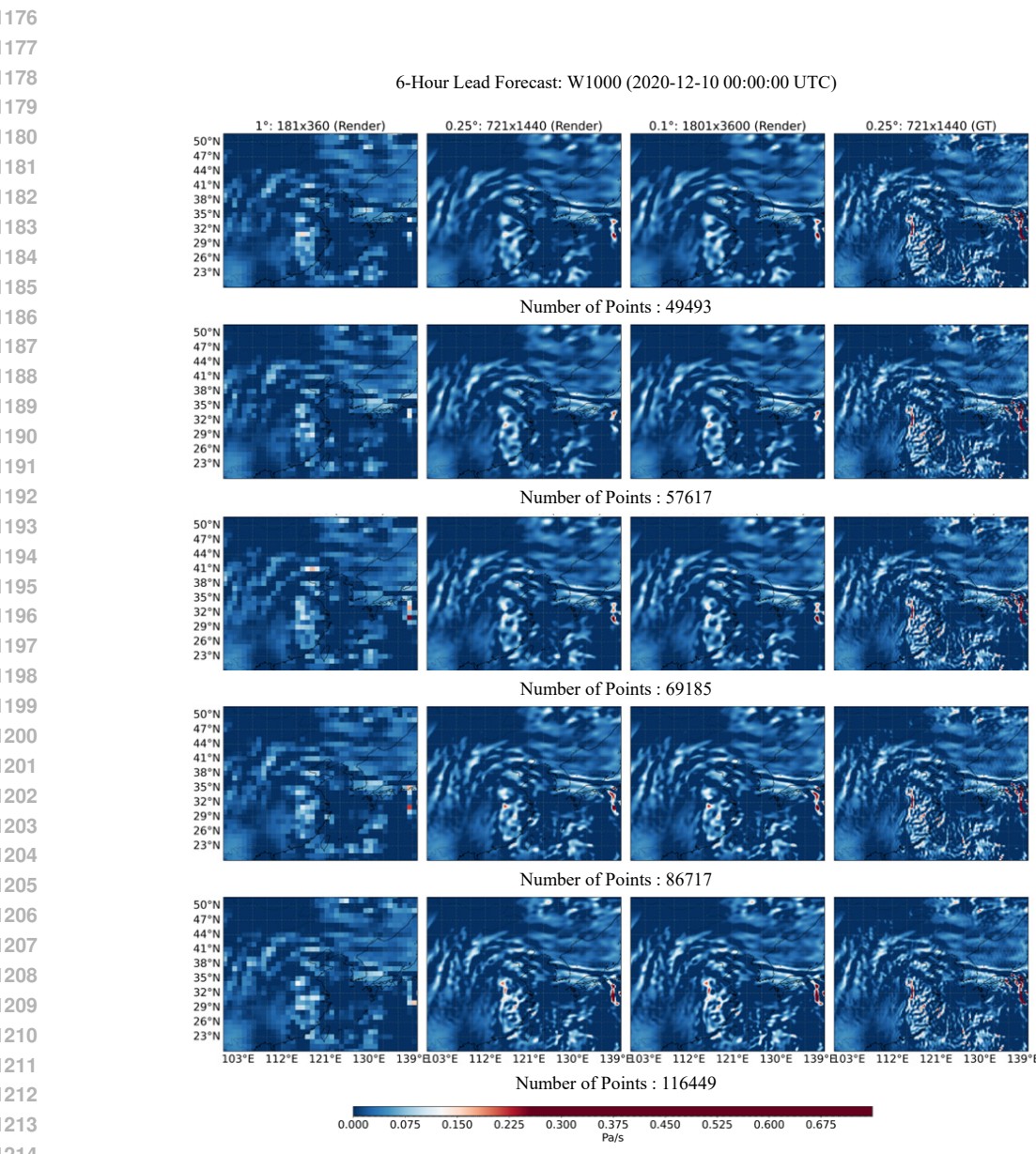

Figure 15: Multi-scale forecasting results of GaussianCast for vertical velocity (W) at the 1000 hPa level at 1° (181×360), 0.25° (721×1440), and 0.1° (1801×3600) resolutions with varying numbers of Gaussian points. While trained on 0.25° ERA5 ground truth data, the model uncovers more coherent vertical motion features at 0.1° resolution with increasing point count.

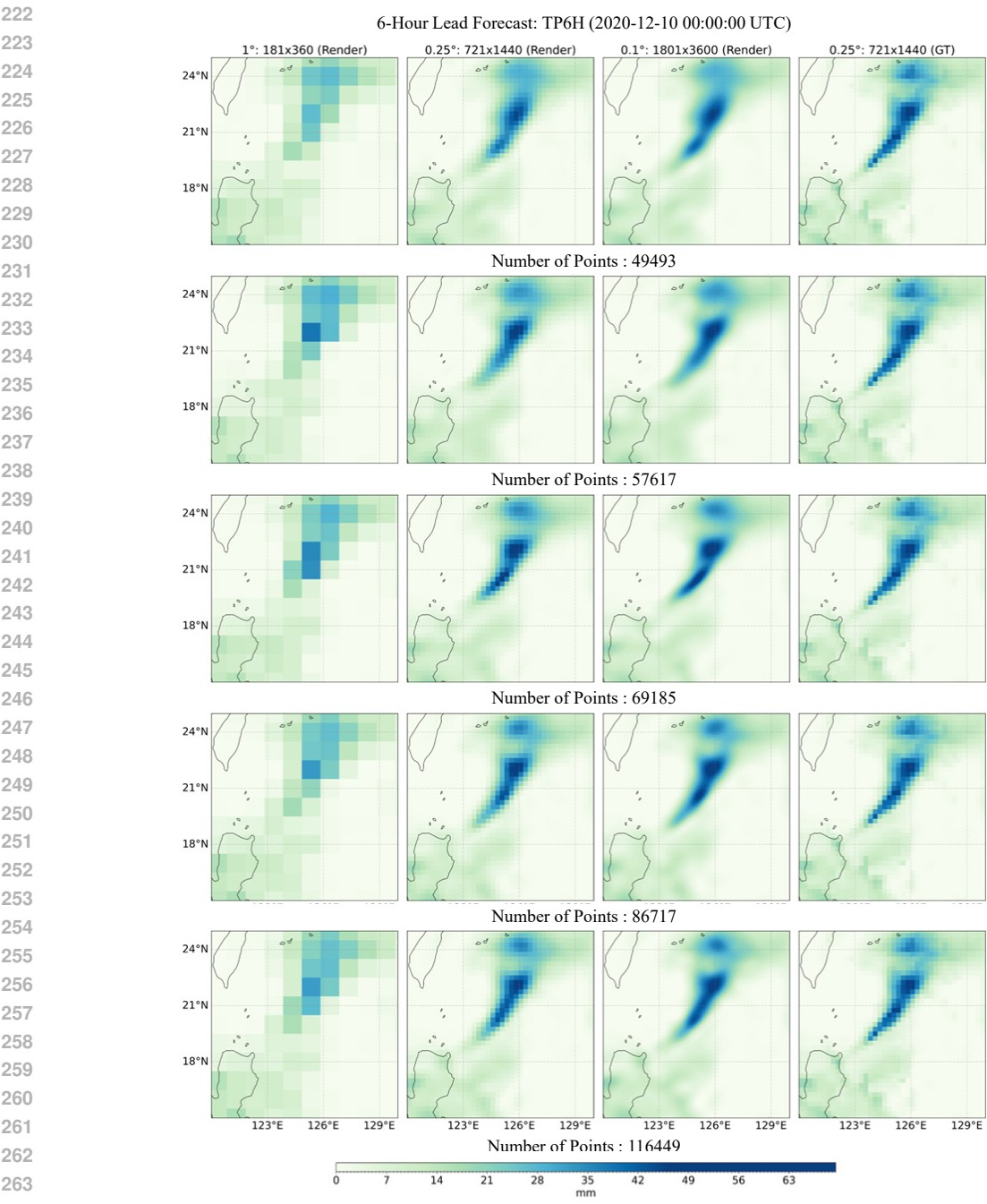

Figure 16: Multi-scale forecasting results of GaussianCast for total precipitation over 6 hours (TP6H) at the surface level at 1° (181×360), 0.25° (721×1440), and 0.1° (1801×3600) resolutions with varying numbers of Gaussian points. While trained on 0.25° ERA5 ground truth data, the model captures sharper precipitation patterns and localized convective events as point density increases.

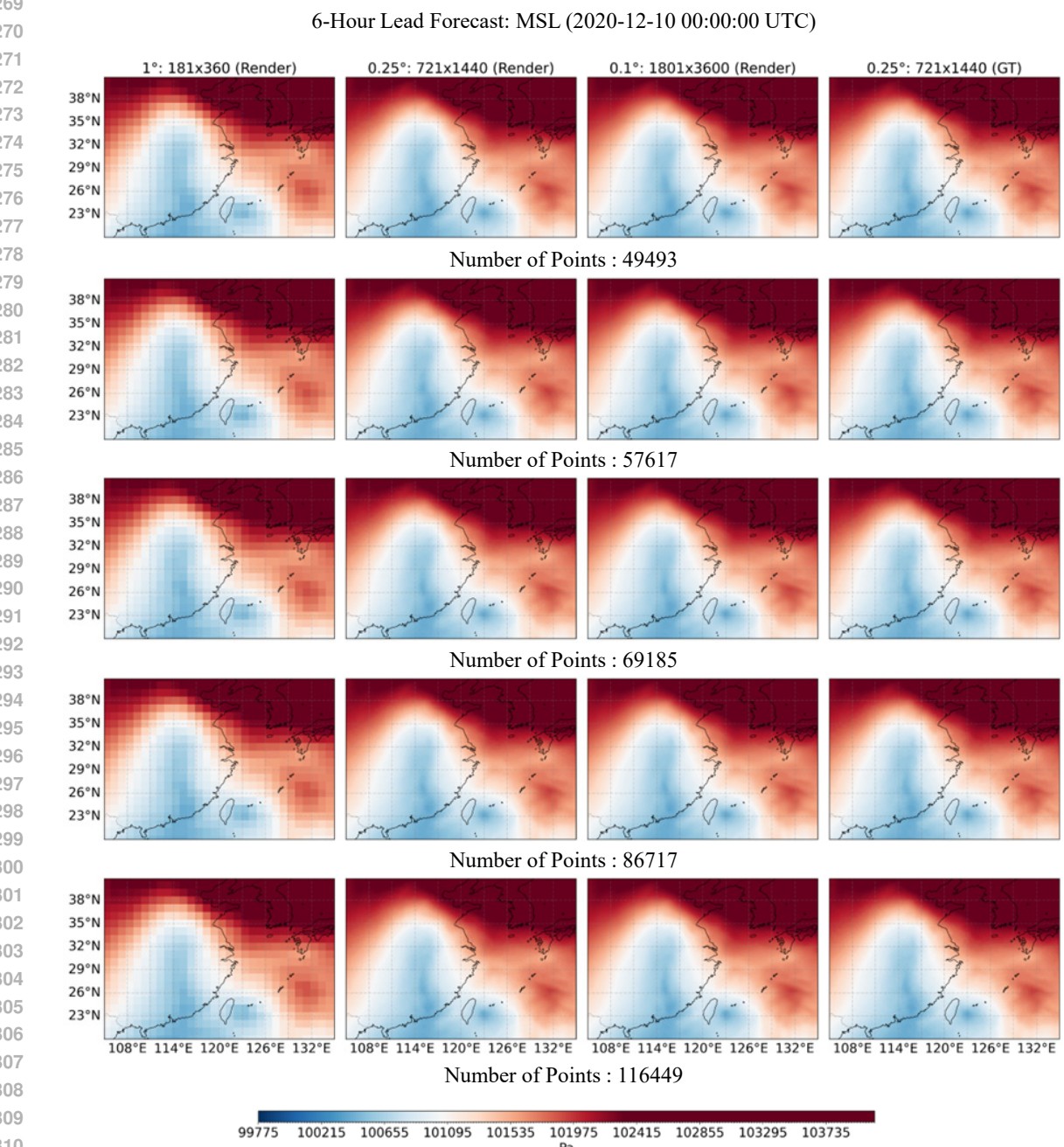

Figure 17: Multi-scale forecasting results of GaussianCast for mean sea level pressure (MSL) at 1° (181×360), 0.25° (721×1440), and 0.1° (1801×3600) resolutions with varying numbers of Gaussian points. While trained on 0.25° ERA5 ground truth data, the model maintains physically consistent pressure fields and improves system boundary clarity at finer resolutions.

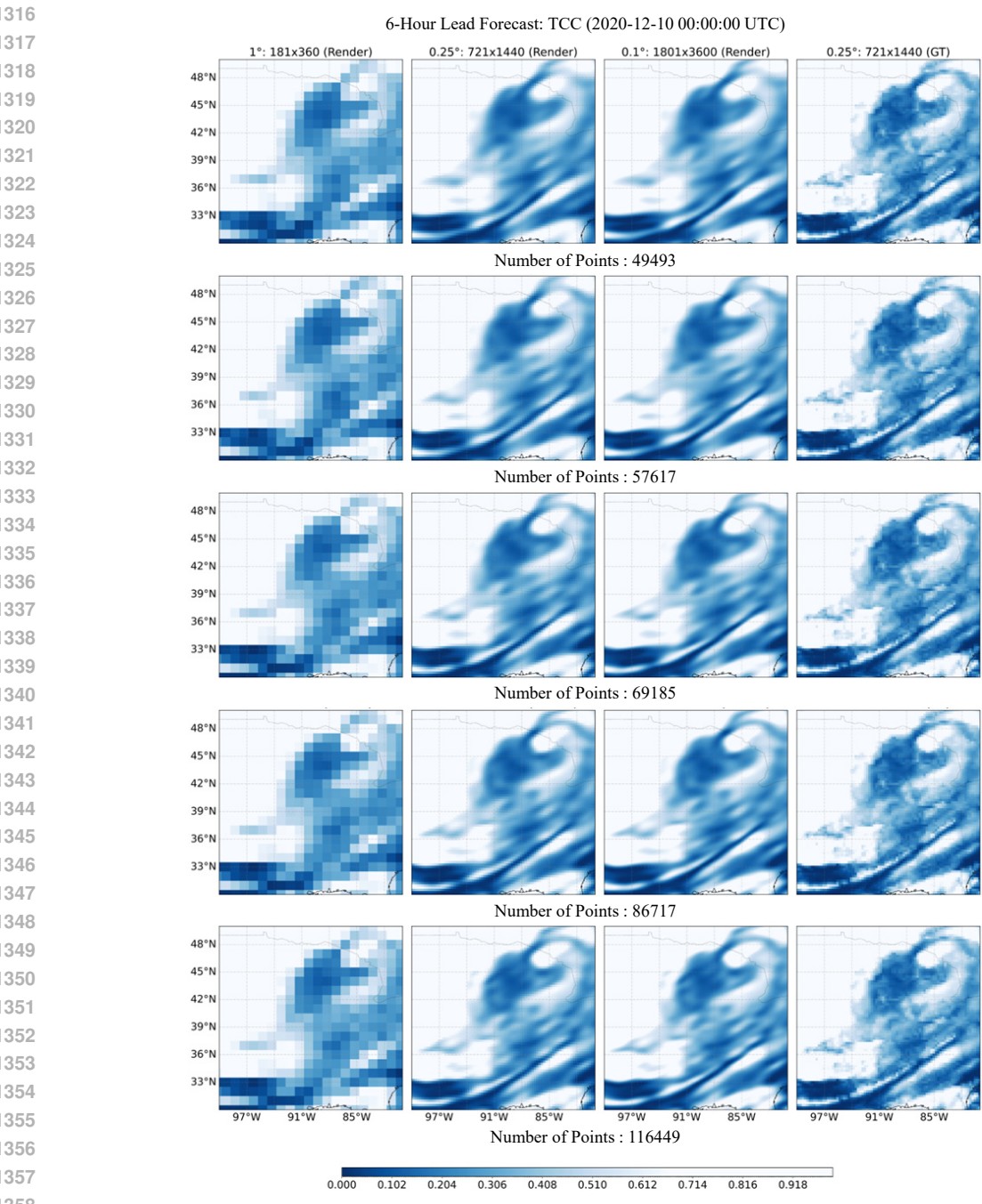

Figure 18: Multi-scale forecasting results of GaussianCast for total cloud cover (TCC) at the surface level at 1° (181×360), 0.25° (721×1440), and 0.1° (1801×3600) resolutions with varying numbers of Gaussian points. While trained on 0.25° ERA5 ground truth data, the model refines cloud structure predictions at 0.1° resolution with increasing spatial density.

