# OpenReview forum: "Breaking the Gridlock: Efficient Atmospheric Data Reconstruction and Prediction via Generative 3D Gaussian Splatting"
_ICLR.cc/2026/Conference — Submitted to ICLR 2026_

### Official Review · Reviewer_ZQTg · 2025-10-26

**Soundness:** 3
**Presentation:** 3
**Contribution:** 2
**Rating:** 4
**Confidence:** 4

**Summary:**

This paper presents a novel, computationally lightweight approach for medium-range forecasting based on applying 3D Gaussian Splatting. The approach safes an order of magnitude inference compute compared to models like GraphCast, and achieves reasonable performance scores.

**Strengths:**

The idea of 3D Gaussian Splatting is original, it leads to high computational efficiency while maintaining performance to a certain level.

The work supports multi resolution out of the box.

The proposed method is sufficiently benchmarked within the weatherbench framework to get an initiation on strength and weaknesses.

**Weaknesses:**

The main claim of this work is to increase computational efficiency while maintaining performance. This should be investigated in more detail:
- AI emulators have demonstrated 10000x speed ups compared to NWP — this work shows 10x speed up over AI emulators — what does that mean? What can I do that I could not do before? Is it beneficial to accept the lowered accuracy for an enhanced computational efficiency?
- FLOPs are the only measurement used, why not wallclock time etc? At the moment, it’s hard to understand what the actual benefit of using this model really is besides that it’s grounded in a good idea and seems to require less FLOPs.


Overall, I don’t see significant contributions besides the speed ups which are not investigated in detail.

**Questions:**

How does a simple learned MLP mapping from RGG to lat-lon compare to the full 3DGS pipeline? This would justify the architectural complexity.


So-what question in my head is: Is it justified to assume a worse method that runs quicker will be used (keeping in mind that other emulators already speed up NWP by 10,000x)?

---

> ### Author Response · Authors · 2025-11-22
>
> We thank the reviewer for the valuable feedback.
>
> **Response to weakness1.1**: The reviewer mentioned a 10,000× acceleration of AI emulators over traditional NWP. However, widely-used AI forecast models such as GraphCast, Pangu-Weather, FengWu, and FuXi **typically require 14–60 seconds of inference to generate a 10-day global forecast (6-hour steps)**, whereas the ECMWF IFS-HRES system requires **approximately 1 hour** for the same task. This corresponds to a realistic **speed-up of about 60–257×**, rather than 10,000×.
> We provide in the table below the training time, inference time, and GPU configuration used by each model reported in the literature. This allows a direct and transparent comparison of computational efficiency across representative AI weather forecasting systems.
> |Model|Training Time|Training device|Inference Time for 10-day Forecast (6-hour steps)|Inference device|
> |-|-|-|-|-|
> |IFS-HRES (ECMWF)|N/A (operational NWP system)|N/A|~1 hour|N/A|
> |GraphCast|4 weeks|32 TPUv4 |60 seconds|1 TPUv4|
> |Pangu-Weather|15 days|192 Nvidia V100 GPUs|14 seconds|1 Nvidia V100 GPU|
> |FengWu|17days|32 Nvidia A100 GPUs|30 seconds|1 Nvidia A100 GPU|
> |FuXi|Pre-training 30 hours; Fine-tune 6 days|8 Nvidia A100 GPUs|50 seconds|1 Nvidia A100 GPU|
> |Our Model|3 days|8 Nvidia A100 GPUs|10 seconds|1 Nvidia A100 GPU|
>
> **Response to weakness1.2**: A acceleration over existing AI emulators significantly improves computational efficiency, **especially in the context of ensemble forecasting.** Although single-run AI inference is fast, **generating a 50-member ensemble with existing models still requires 12–50 minutes,** because each member must be inferred independently. This remains a bottleneck for operational probabilistic forecasting. A model that is an order of magnitude more efficient substantially lowers the total cost of ensemble generation and **makes large-scale ensembles (hundreds or even thousands of members) practically achievable.** This is also why systems such as NVIDIA’s FourCastNetV3 continue to focus on optimization for large scale ensemble prediction.
>
> **Response to weakness1.3 and question2**: Importantly, our model **does not trade accuracy for efficiency.** The goal of this work is to maintain forecast skill comparable to existing AI emulators while significantly improving both training and inference efficiency. Our experiments show that the model preserves **competitive performance relative to established baselines.**
>
> **Response to weakness2 and question2**: We report actual training and inference times in the table above. These results show that our model significantly reduces both training and inference time compared to baselines. Moreover, by using the generative 3DGS method on discretized atmospheric grids, our model can **reconstruct full 3D atmospheric fields and efficiently sample weather data at arbitrary resolutions from continuous Gaussian fields. This enables not only faster computation but also flexible and high-resolution data generation, which goes beyond simple FLOP reduction.**
>
> **Response to question1**: A simple learned MLP **cannot achieve the desired mapping from RGG to latitude-longitude grids.** The relationship between the irregular RGG representation and the structured lat-lon grid is highly non-linear and spatially complex, which cannot be captured by a straightforward MLP.

---

### Official Review · Reviewer_X4vd · 2025-10-27

**Soundness:** 3
**Presentation:** 3
**Contribution:** 2
**Rating:** 4
**Confidence:** 5

**Summary:**

This paper introduces GaussianCast, a novel framework for numerical weather prediction. It overcomes the data redundancy of traditional latitude-longitude grids by combining 3D Gaussian Splatting (3DGS) with a Reduced Gaussian Grid (RGG). A generative model, using multi-scale Graph Attention Transformers, is conditioned on the current atmospheric state to predict the parameters (covariance, features, opacity) of 3D Gaussians representing the future global atmospheric field.

**Strengths:**

This paper, "Breaking the Gridlock: Efficient Atmospheric Forecasting with 3D Gaussian Splatting on Reduced Gaussian Grids," presents several compelling strengths that make it a solid and valuable contribution.

1.  **Exceptional Clarity and Readability:** The paper is exceptionally well-written and structured. The narrative flows logically from the problem statement (redundancy in latitude-longitude grids) to the core insight (using 3DGS for continuous representation) and the solution (GaussianCast).
2.  **Rigorous and Exhaustive Analysis:** The experimental section is thorough and convincing.
3.  **Impressive and Well-Demonstrated Model Efficiency:** This is arguably the paper's most significant contribution. The results on efficiency are not just claimed but are robustly supported by data. Achieving competitive performance with state-of-the-art models like GraphCast and Pangu-Weather while using **orders of magnitude less computational resources** (101M parameters vs. Billions, 1843 GFLOPs vs. ~20,000 GFLOPs) is a remarkable feat. The ability to generate a 10-day forecast in just 20 seconds on a single A100 GPU highlights its practical potential for operational settings with limited resources.

**Weaknesses:**

### **Major Points**

1.  **Novelty and Atmospheric Physics Integration:** The two core components of this work—3D Gaussian Splatting and Graph Attention mechanisms—are well-established techniques borrowed from other fields. A significant concern is the lack of demonstrable, novel design specifically tailored to the unique properties and physics of atmospheric data. This limits the novelty of the contribution.

2.  **Performance Scaling and SOTA Comparison:** The results in Table 1 show that the current model lags behind state-of-the-art (SOTA) baselines across all metrics. While the computational efficiency is commendable, it is crucial to provide a more detailed analysis of performance scaling. Specifically, if the model's parameters and computational budget were scaled up to the level of other SOTA models (e.g., ~10^4 GFLOPs), could its performance surpass theirs? Figure 7 partially addresses scaling with RGG points, but a direct comparison against other baselines under increased capacity is needed.

3.  **Ablation Study Completeness:** The model is primarily composed of the 3DGS representation and the GAT backbone. A critical ablation is missing: what is the performance if 3DGS is removed and the GAT is trained in a more conventional manner, with input and output operating directly on a uniform grid (or graph thereof), similar to models like GraphCast? This experiment is necessary to isolate the specific contribution of the 3DGS component to the overall performance.

4.  **Physical Consistency and Information Loss:** The use of a compressed, non-uniform grid representation (RGG) raises questions about potential loss of physical information. The authors should explicitly address this concern. For instance, a more convincing demonstration would be to interpolate both the compressed data and the original 721x1440 data to a common, higher-resolution uniform grid for comparison, ensuring that fine-scale physical structures are preserved.

5.  **Comparison with Contemporary Models:** The paper lacks comparisons with several recent and relevant atmospheric forecasting models, particularly CirT (https://arxiv.org/abs/2502.19750) and OneForecast (https://arxiv.org/abs/2502.00338). Given that the multi-scale attention and graph-based approach bears significant resemblance to models like OneForecast and GraphCast, their omission weakens the contextualization of this work within the current literature.

### **Minor Points**

6.  **Clarity on Grid Usage:** It should be clarified whether the "reduced Gaussian grid" used in this work is the same as or different from the grid used in GraphCast.

7.  **High-Resolution Inference Details (Section 4.5):** The process of training on 0.25° data and performing inference at 0.1° resolution is a key feature. However, the methodological details of how this is achieved are insufficiently explained and need elaboration.

8.  **Mathematical Typos:**
    *   Line 140: The expression for the Gaussian center coordinate `μ_i` seems to have a typographical error (likely a missing comma).
    *   Line 188: The dimensionality notation `p_i ∈ R^F` is inconsistent and should presumably be `p_i ∈ R^K` based on the defined frequency bands.

9.  **Appendix Figure Quality:** The image quality in the appendix is poor and should be replaced with high-resolution versions for clarity.

**Questions:**

1.  **Novelty & Physical Integration:** Beyond the application of existing techniques (3DGS, GAT), what is the specific *novel methodological contribution* of this work to atmospheric science? Could you clarify how the design explicitly incorporates and benefits from the unique properties of atmospheric physics, rather than being a direct transfer of models from other domains?

2.  **Performance Ceiling & Scaling:** The results indicate the model does not surpass SOTA performance despite high efficiency. What is the performance ceiling of your proposed architecture? If scaled up in parameters and computational budget (e.g., to ~10^4 GFLOPs), could it outperform models like GraphCast or FengWu? Please provide a scaling analysis or discussion on this point.

3.  **Ablation of Core Components:** To isolate the contribution of the 3DGS representation, what would be the performance of an ablated model that uses only the GAT backbone for direct prediction on a uniform grid (similar to GraphCast), keeping all other factors (number of parameters, training data) comparable?

4.  **Physical Fidelity & Grid Comparison:**
    *   Does the compression and representation on the Reduced Gaussian Grid lead to a loss of fine-scale physical information? Please provide a comparative analysis (e.g., by interpolating both your compressed data and the original data to a common high-resolution grid) to demonstrate the preservation of physical structures.
    *   How does your grid methodology specifically differ from that used in GraphCast?
    *   Furthermore, how does your method compare quantitatively with other recent graph-based models like OneForecast and CirT, which share conceptual similarities?

---

> ### Author Response · Authors · 2025-11-18
>
> We thank the reviewer for the valuable feedback.
>
> **Response to weakness1(W1) and question1(Q1):** Conventional 3DGS optimizes Gaussian parameters **independently per sample and cannot generalize.** We instead propose a conditional generative 3DGS that **learns a shared mapping from atmospheric states to Gaussian parameters, converting 3DGS from a reconstruction tool into a forecasting capable continuous representation.** We further extend 3DGS from **RGB to >160 atmospheric variables, redesign its Gaussian features, and replace COLMAP with a physics-aware initialization using RGG lat–lon coordinates for meaningful spatial alignment.** Finally, the learned continuous Gaussians enables **resolution-free sampling and compresses the native 721×1440(~1 million) grid points to 69k Gaussians, offering both computational efficiency and flexible multi-scale modeling, capabilities absent in existing atmospheric prediction methods.** Overall, the novelty of our work does not lie in using GAT, but in reformulating grid-based atmospheric fields as continuous 3D Gaussian distributions through a new conditional generative mechanism and physics-aware design.
>
> **Response to W2 and Q2:** As shown in §4.4, **increasing the number of Gaussians consistently reduces reconstruction error**, indicating that the model has not reached its capacity limit. We conducted additional forecasting experiments using 116449 Gaussian points (comparable to the original grid size). The experimental results will be provided later, as the model is currently being trained.
>
> **Response to W3 and Q3:** The 3DGS component is essential because **it converts the discrete grid (~1 million) into a compact continuous Gaussian field (69k points),** which both reduces computational cost and enables **resolution-free sampling.** A GAT operating directly on the full grid is computationally prohibitive and **cannot generalize across resolutions.** Nevertheless, we agree that this ablation is informative. The experimental results will be provided later, as the model is currently being trained.
>
> **Response to W4,6 and Q4:** The conventional lat–lon grid used in models such as GraphCast assigns **equal longitudinal resolution at all latitudes**, which is **physically inconsistent because meridians converge toward the poles.** The Reduced Gaussian Grid (RGG) addresses this by **reducing redundant polar points and allocating fewer samples at high latitudes and more near the equator,** better matching Earth’s spherical geometry.
> In our method, **RGG is used only to initialize the centers of the 3D Gaussians**. The model then learns all Gaussian parameters to form a continuous, physically aligned Gaussian field **(denser at the equator, sparser at the poles).** Because the Gaussian representation is continuous, the compression introduces minimal physical information loss, and the learned field still respects large-scale and fine-scale spatial structures.
> We quantitatively compare high-resolution upscaling from a 721×1440 grid to 2001×4000, evaluating the reconstruction RMSE of a continuous Gaussian field against interpolation:
> |Channel|z500|t850|q700|wv850|t2m|u10|v10|msl|
> |-|-|-|-|-|-|-|-|-|
> |Interpolation|7.73|0.36|0.21|0.72|0.81| 0.64|0.65|36.46|
> | Ours | **6.07** | **0.29** | **0.15** | **0.63** | **0.68** | **0.56** | **0.51** | **35.17** |
>
>
> **Response to W5 and Q4:** CIRT targets ≥3-week forecasts, whereas ours is medium-range. For completeness, we report OneForecast results:
> | Channel | z_500 | u_10 | v_10 | wind_850 |
> |-|-|-|-|-|
> || 1day / 3day / 5day / 7day | 1day / 3day / 5day / 7day | 1day / 3day / 5day / 7day | 1day / 3day / 5day / 7day |
> |OneForecast|43 / 139 / 299 / 502|0.80 / 1.82 / 2.78 / 3.68|0.74 / 1.53 / **2.38** / 3.09| 1.99 / 3.05 / 4.7 / 6.2 |
> | ours | **40 / 126 / 272 / 441** | **- / 1.46 / 2.28 / -** | **- / 1.50 /** 2.46 / - | **1.53 / 2.84 / 4.32 / 5.76** |
>
> **Response to W7:** Our approach uses the continuous 3D Gaussian field to enable inference at resolutions finer than the training grid. The model is trained on 0.25° data, reconstructing the atmospheric field as a continuous 3D Gaussian distribution. Each Gaussian has a feature $f_i$ (representing atmospheric variables) and an opacity $\alpha_i$ that determines its contribution to any spatial location. During inference, we evaluate the sum of Gaussian contributions at a desired 0.1° high-resolution grid, effectively “rendering” from the continuous field to the finer resolution. The 0.25° grid provides training supervision, while the continuous Gaussian representation allows arbitrary-resolution sampling of the reconstructed atmospheric field, preserving the learned structures without retraining. This process is formalized in §3.3, Eq.6, where the grid values are computed as weighted sums of Gaussian contributions.
>
> **Response to W8 and W9:** We correct these mathematical typos immediately. Appendix images are high-res; please allow time for loading or use other PDF viewer.

---

> ### Author Response · Authors · 2025-12-03
>
> **Response to W2 and Q2:** We appreciate the reviewers' suggestions. The previously ongoing experiments (using 116,449 Gaussian points, comparable to the original grid size) have now been fully trained and completed. As shown in Table 1 and Table 2, **increasing the number of Gaussians continues to reduce forecasting error (RMSE), confirming that the model has not reached its capacity limit. These results are consistent with our findings in §4.4.**
>
> **Table 1. Effect of Increasing the Number of Gaussians (for W2/Q2). The RMSE of upper-levelvariables.**
> | Channel | Gaussians | 1 day | 3 days | 5 days | 7 days |
> |-|-|-|-|-|-|
> |z500 | 69185    | 40 | 126 | 272 | 441 |
> |z500 | 116449     | 36 | 116  | 250 | 365 |
> |t850 | 69185   | 0.58 | 0.98| 1.49 |2.16|
> |t850 |  116449     | 0.50 | 0.92   | 1.47|2.09|
> |wv850 | 69185     | 1.53 | 2.84 | 4.32 | 5.76|
> |wv850 |  116449 |  1.40 | 2.61 |4.14 |5.51|
>
> **Table 2. Effect of Increasing the Number of Gaussians (for W2/Q2).  The RMSE of Surface-level variables.**
> | Channel | Gaussians | 6 h| 72h | 120h |
> |-|-|-|-|-|
> |T2M | 69185     |0.56 | 0.98 | 1.45|
> |T2M |  116449  |0.54 | 0.92| 1.31 |
> |U10 | 69185     | 0.42| 1.46 | 2.28 |
> |U10 |  116449   |0.37 | 1.40 | 2.26 |
> |V10 | 69185     | 0.44 | 1.50 | 2.46 |
> |V10 |  116449   |0.45 | 1.45  | 2.40|
> |MSL| 69185     | 25.4 | 138.4 | 279.8 |
> |MSL|  116449  | 24.1| 132.5 | 269.4 |
>
>
> **Response to W3 and Q3**
> The additional ablation requested by the reviewers has also been completed. The newly finished experiments directly compare: (1) a GAT operating on the full discrete grid (~1M points), and (2) our 3DGS-based compressed representation (69k points). As summarized in Table 3, operating directly on the full grid leads to substantially **higher memory usage and computational cost, while achieving performance that is only comparable to our 3DGS-based approach.** This confirms that the 3DGS component provides a significantly more **efficient representation without sacrificing accuracy, and is therefore essential for both scalability and generalization across resolutions.**
>
> **Table 3. Ablation: GAT on Full Grid vs. 3DGS (for W3/Q3). The RMSE of upper-levelvariables.**
> | Channel | Gaussians | 1 day | 3 days | 5 days | 7 days |
> |-|-|-|-|-|-|
> | z500 | 3DGS + GAT (ours) | 40 | 126 | 272 | 441 |
> | z500 | GAT on Full Grid | 38 | 125 | 275 | 459 |
> | t850 | 3DGS + GAT (ours) | 0.58 | 0.98| 1.49 |2.16 |
> |t850 | GAT on Full Grid | 0.55| 0.97  | 1.53|2.27|
> |wv850 | 3DGS + GAT (ours) | 1.53 | 2.84 | 4.32 | 5.76|
> |wv850 | GAT on Full Grid| 1.49 | 2.73  | 4.48 |5.93|
>
>
> **Table 4. Ablation: GAT on Full Grid vs. 3DGS (for W3/Q3).  The RMSE of Surface-level variables.**
> | Channel | Model | 6 h| 72h | 120h |
> |-|-|-|-|-|
> |T2M |  3DGS + GAT (ours)  |0.56 | 0.98 | 1.45|
> |T2M |  GAT on Full Grid | 0.50| 0.96 | 1.39|
> |U10 | 3DGS + GAT (ours) | 0.42| 1.46 | 2.28 |
> |U10 |  GAT on Full Grid | 0.40 | 1.49 | 2.34 |
> |V10 | 3DGS + GAT (ours) | 0.44 | 1.50 | 2.46 |
> |V10 |  GAT on Full Grid | 0.46 |1.47 | 2.43|
> |MSL| 3DGS + GAT (ours) | 25.4 | 138.4 | 279.8 |
> |MSL|  GAT on Full Grid | 24.3| 136.5 |278.6 |

---

### Official Review · Reviewer_zvGn · 2025-11-01

**Soundness:** 2
**Presentation:** 3
**Contribution:** 3
**Rating:** 4
**Confidence:** 4

**Summary:**

GaussianCast proposes a novel framework based on generative 3D Gaussian splatting (3DGS) for compact, continuous representation and efficient prediction of high-dimensional atmospheric data. The core of the model places 3DGS Gaussian centers on a Reduced Gaussian Grid (RGG) to achieve equal-area sampling and up to 14x data compression. Using a multi-scale Graph Attention Transformer (GAT), the model generates the Gaussian parameters (covariance, feature vectors, and opacity) for the next time step based on the current atmospheric state, enabling 6-hour interval global medium-range weather forecasting through differentiable rasterization. GaussianCast can generate a 10-day global forecast within 20 seconds, demonstrating the potential of 3DGS in data-driven medium-range weather prediction.

**Strengths:**

- **Methodological Innovation and Potential:** This work is the first to introduce generative 3DGS into the field of numerical weather prediction, combining it with RGG to address data redundancy in traditional latitude-longitude grids. This provides an efficient and promising new direction for continuous representation and prediction of high-dimensional atmospheric fields.

- **Technical Soundness:** The model design, particularly the integration of the RGG structure into 3DGS and the use of multi-scale GATs to handle irregular graph structures and capture global teleconnections, demonstrates solid technical implementation. The paper is well-structured and clearly presented.

- **Efficiency and Compression Advantages:** The model achieves up to 14x data compression and significantly reduces computational cost while maintaining performance, which is crucial for future scaling to higher-resolution weather forecasting.

**Weaknesses:**

- **Insufficient and Partial Presentation of Experimental Results:**

    - **Limited Variable Coverage:** While the model aims to predict 160 atmospheric variables, the core experiments (e.g., Table 1) primarily highlight forecasting performance for only four upper-air variables (Z500, T850, Q700, V850) and a few near-surface variables. A comprehensive evaluation and presentation of performance across the other hundred-plus variables (e.g., humidity, divergence at different pressure levels) are lacking, making the claimed "high-dimensional" advantage insufficiently validated.

    - **Limited Visualization:** Although GaussianCast is a global model, the forecast visualizations presented in the paper typically focus on local or regional results, which makes it difficult to fully demonstrate its predictive capability, smoothness, and coherence on a global scale.

- **Limited Research Significance and Performance Improvement:**

    - **Marginal Performance Gains:** From the surface variable forecasting performance shown in Table 1, GaussianCast does not clearly outperform several existing baseline models. Similarly, the tropical cyclone track prediction results in Figure 5 do not show decisive advantages.

    - **Limited Efficiency Advantage:** The claimed speed of generating a 10-day forecast in 20 seconds is not the fastest among current models. For example, FourCastNetV2 with similar resolution can also achieve a 10-day forecast in 20 seconds, and WeatherMesh-3 can complete a 14-day forecast within 10–12 seconds. Therefore, the model's primary contribution lies in methodological exploration rather than leading in absolute performance or speed.

- **Missing Experiments on Methodological Limitations:** Given that the main contribution of this work is the introduction of the 3DGS method, experiments addressing the inherent limitations of 3DGS should be supplemented, such as:

    - Lack of analysis on the model’s ability to handle unseen extreme weather events (generalization).

    - Lack of detailed evaluation on how 3DGS rendering at different sampling rates affects data smoothness or local detail preservation.

**Questions:**

1. Technical Correction Regarding Related Work: Please note that the statement in lines 098–099—“However, these models rely on latitude–longitude grids”—contains a technical inaccuracy. Recent AI forecasting models, such as AIFS (Lang, Simon, et al. "AIFS—ECMWF's data-driven forecasting system." arXiv preprint arXiv:2406.01465 (2024)), which is cited in your related work, already adopt the RGG. Therefore, this statement underestimates the progress made by the current field in addressing latitude-longitude grid redundancy and should be corrected to ensure accuracy.

2. Mechanistic Explanation of Conditional Generation and Generalization: In the introduction (lines 051–053), you correctly identify the inherent generalization weakness of traditional 3DGS (i.e., overfitting to individual samples). You propose that a generative framework conditioned on the "current atmospheric state" can overcome this issue, but a deeper explanation of this mechanism is lacking. Could you clarify: why does simply conditioning the current atmospheric state as input to the GAT effectively mitigate the generalization limitations inherent to 3DGS, which stem from its explicit point cloud representation? What is the theoretical basis behind this?

---

> ### Author Response · Authors · 2025-11-13
>
> We thank the reviewer for the comment. Due to page limits, the main text shows results for a subset of key variables (Z500, T850, Q700, V850, and a few near-surface fields). Here, we provide additional results for the remaining variables at other levels, showing that our model maintains strong accuracy across other variables. Comparison with baseline models can be found in WeatherBench, as space here is limited. Full results will be included in the supplementary material.
>
> | Channel | 6 hour | 24 hour | 60 hour | 120 hour | 180 hour | 240 hour | 300 hour | 360 hour |
> |----------|--------|---------|---------|----------|----------|----------|----------|----------|
> | z_1000 | 19.45 | 43.11 | 90.50 | 226.97 | 364.67 | 462.80 | 528.08 | 558.73 |
> | z_925 | 17.92 | 40.53 | 86.17 | 218.07 | 352.92 | 449.73 | 512.87 | 542.66 |
> | z_850 | 17.19 | 39.29 | 84.01 | 213.39 | 348.34 | 446.86 | 510.17 | 540.84 |
> | z_700 | 16.60 | 39.47 | 87.68 | 225.07 | 375.08 | 488.76 | 562.13 | 599.34 |
> | z_600 | 16.75 | 41.28 | 96.61 | 248.99 | 418.04 | 547.40 | 631.83 | 675.59 |
> | z_400 | 19.28 | 50.60 | 133.17 | 339.63 | 570.71 | 745.15 | 860.64 | 922.30 |
> | z_300 | 21.10 | 59.07 | 151.97 | 382.44 | 651.93 | 856.13 | 993.29 | 1067.45 |
> | z_250 | 22.08 | 63.31 | 150.89 | 371.90 | 648.28 | 862.73 | 1009.59 | 1090.33 |
> | z_200 | 23.59 | 63.86 | 137.47 | 323.71 | 581.76 | 792.43 | 943.71 | 1030.07 |
> | z_150 | 26.01 | 62.29 | 123.66 | 266.75 | 490.23 | 685.66 | 835.69 | 927.45 |
> | z_100 | 30.61 | 63.73 | 121.74 | 231.21 | 420.73 | 597.95 | 743.82 | 839.97 |
> | z_50 | 44.57 | 87.00 | 162.40 | 281.49 | 439.00 | 606.25 | 767.35 | 883.51 |
> | q_1000 | 2.8e-4 | 4.5e-4 | 6.1e-4 | 8.6e-4 | 1.1e-3 | 1.3e-3 | 1.5e-3 | 1.6e-3 |
> | q_925 | 3.7e-4 | 6.3e-4 | 8.2e-4 | 1.0e-3 | 1.3e-3 | 1.4e-3 | 1.6e-3 | 1.7e-3 |
> | q_850 | 4.0e-4 | 7.8e-4 | 1.1e-3 | 1.4e-3 | 1.6e-3 | 1.7e-3 | 1.9e-3 | 2.0e-3 |
> | q_600 | 2.1e-4 | 4.6e-4 | 6.5e-4 | 8.7e-4 | 1.0e-3 | 1.1e-3 | 1.3e-3 | 1.3e-3 |
> | q_500 | 1.5e-4 | 3.3e-4 | 4.6e-4 | 6.0e-4 | 7.0e-4 | 7.7e-4 | 8.6e-4 | 9.0e-4 |
> | q_400 | 8.0e-5 | 1.8e-4 | 2.4e-4 | 3.1e-4 | 3.5e-4 | 3.9e-4 | 4.4e-4 | 4.6e-4 |
> | q_300 | 3.0e-5 | 6.0e-5 | 8.0e-5 | 1.0e-4 | 1.1e-4 | 1.2e-4 | 1.4e-4 | 1.5e-4 |
> | q_250 | 1.0e-5 | 3.0e-5 | 4.0e-5 | 4.0e-5 | 5.0e-5 | 5.0e-5 | 6.0e-5 | 6.0e-5 |
> | q_200 | 0.0 | 1.0e-5 | 1.0e-5 | 1.0e-5 | 1.0e-5 | 1.0e-5 | 2.0e-5 | 2.0e-5 |
> | q_150 | 0.0 | 0.0 | 0.0 | 0.0 | 0.0 | 0.0 | 0.0 | 0.0 |
> | q_100 | 0.0 | 0.0 | 0.0 | 0.0 | 0.0 | 0.0 | 0.0 | 0.0 |
> | q_50 | 0.0 | 0.0 | 0.0 | 0.0 | 0.0 | 0.0 | 0.0 | 0.0 |
> | u_1000 | 0.46 | 0.94 | 1.50 | 2.55 | 3.40 | 3.92 | 4.32 | 4.46 |
> | u_925 | 0.54 | 1.18 | 1.92 | 3.30 | 4.39 | 5.04 | 5.51 | 5.68 |
> | u_700 | 0.65 | 1.39 | 2.13 | 3.51 | 4.73 | 5.57 | 6.16 | 6.47 |
> | u_600 | 0.65 | 1.47 | 2.30 | 3.85 | 5.24 | 6.19 | 6.87 | 7.22 |
> | u_500 | 0.71 | 1.59 | 2.60 | 4.44 | 6.05 | 7.17 | 7.94 | 8.34 |
> | u_400 | 0.81 | 1.80 | 3.06 | 5.35 | 7.32 | 8.66 | 9.59 | 10.06 |
> | u_300 | 0.88 | 1.98 | 3.44 | 6.23 | 8.64 | 10.30 | 11.49 | 12.10 |
> | u_250 | 0.90 | 2.03 | 3.41 | 6.12 | 8.66 | 10.47 | 11.83 | 12.55 |
> | u_200 | 0.90 | 2.00 | 3.09 | 5.25 | 7.55 | 9.39 | 10.82 | 11.64 |
> | u_150 | 0.88 | 1.95 | 2.70 | 4.11 | 5.81 | 7.38 | 8.66 | 9.54 |
> | u_100 | 0.83 | 1.73 | 2.31 | 3.20 | 4.30 | 5.42 | 6.33 | 6.99 |
> | u_50 | 0.88 | 1.72 | 2.31 | 2.94 | 3.55 | 4.26 | 4.95 | 5.42 |
> | v_1000 | 0.45 | 0.98 | 1.55 | 2.65 | 3.58 | 4.14 | 4.50 | 4.61 |
> | v_925 | 0.54 | 1.21 | 1.96 | 3.39 | 4.53 | 5.18 | 5.59 | 5.73 |
> | v_700 | 0.64 | 1.39 | 2.12 | 3.55 | 4.79 | 5.59 | 6.05 | 6.24 |
> | v_600 | 0.65 | 1.47 | 2.31 | 3.96 | 5.39 | 6.34 | 6.87 | 7.10 |
> | v_500 | 0.70 | 1.61 | 2.63 | 4.63 | 6.33 | 7.45 | 8.08 | 8.33 |
> | v_400 | 0.81 | 1.82 | 3.12 | 5.65 | 7.74 | 9.12 | 9.87 | 10.17 |
> | v_300 | 0.88 | 2.00 | 3.51 | 6.59 | 9.22 | 10.97 | 11.93 | 12.30 |
> | v_250 | 0.89 | 2.05 | 3.45 | 6.45 | 9.26 | 11.19 | 12.29 | 12.72 |
> | v_200 | 0.88 | 2.00 | 3.11 | 5.48 | 8.03 | 9.96 | 11.11 | 11.59 |
> | v_150 | 0.87 | 1.94 | 2.69 | 4.19 | 6.07 | 7.64 | 8.64 | 9.12 |
> | v_100 | 0.84 | 1.74 | 2.25 | 3.13 | 4.34 | 5.48 | 6.24 | 6.62 |
> | v_50 | 0.96 | 1.71 | 2.12 | 2.57 | 3.21 | 3.95 | 4.52 | 4.83 |
> | t_1000 | 0.38 | 0.64 | 0.92 | 1.47 | 2.01 | 2.42 | 2.83 | 3.07 |
> | t_925 | 0.35 | 0.62 | 0.95 | 1.62 | 2.27 | 2.76 | 3.20 | 3.43 |
> | t_700 | 0.28 | 0.50 | 0.81 | 1.49 | 2.17 | 2.68 | 3.06 | 3.25 |
> | t_600 | 0.25 | 0.47 | 0.80 | 1.48 | 2.16 | 2.64 | 3.00 | 3.18 |
> | t_500 | 0.24 | 0.45 | 0.79 | 1.48 | 2.16 | 2.64 | 2.98 | 3.16 |
> | t_400 | 0.24 | 0.45 | 0.77 | 1.41 | 2.06 | 2.53 | 2.87 | 3.06 |
> | t_300 | 0.22 | 0.47 | 0.77 | 1.31 | 1.78 | 2.14 | 2.45 | 2.64 |
> | t_250 | 0.23 | 0.49 | 0.85 | 1.58 | 2.12 | 2.47 | 2.74 | 2.90 |
> | t_200 | 0.27 | 0.52 | 0.87 | 1.71 | 2.46 | 2.96 | 3.29 | 3.49 |
> | t_150 | 0.31 | 0.54 | 0.78 | 1.34 | 1.94 | 2.40 | 2.74 | 2.93 |
> | t_100 | 0.43 | 0.73 | 0.96 | 1.37 | 1.81 | 2.24 | 2.59 | 2.84 |
> | t_50 | 0.59 | 0.91 | 1.20 | 1.52 | 1.72 | 1.99 | 2.26 | 2.51 |

---

> ### Author Response · Authors · 2025-11-13
>
> We thank the reviewer for the valuable feedback.
>
> **Response to weakness 1:**
> GaussianCast’s global forecasting capability is quantitatively demonstrated by the RMSE results, which already capture the model’s **global predictive consistency**. Due to space limitations and image upload constraints, we focused on local visualizations to highlight GaussianCast’s ability to capture fine-grained structures and extreme weather events.
> To address this concern, we will include comprehensive global visualizations in the appendix to better illustrate the model’s global smoothness and coherence.
>
> **Response to weakness 2:**
> Performance improvement: While GaussianCast does not substantially surpass existing SOTA models in raw metrics, our contribution is introducing the 3D Gaussian Splatting (3DGS) paradigm into atmospheric field reconstruction. GaussianCast compresses the global atmospheric grid from **721×1440 (~1 million points) into 69,185 Gaussian points**, achieving strong accuracy while greatly reducing data dimensionality and redundancy.
>
> Efficiency and scalability: Rather than pursuing the fastest inference, **GaussianCast leverages a continuous 3D Gaussian representation that enables both computational efficiency and resolution flexibility. By modeling the atmosphere as a continuous field rather than a fixed grid, GaussianCast can extrapolate reconstructions and forecasts to arbitrary resolutions without retraining, the grid-based models lack.** GaussianCast’s significance lies in transforming discrete grid-based forecasting into a continuous, resolution-agnostic generative process, while maintaining competitive accuracy and efficiency.
> Additionally, WeatherMesh-3 and the ECMWF reproduction of FourCastNet (FourCastNetV2) have not released quantitative RMSE results,. We include the officially released FourCastNet results below, where GaussianCast achieves lower global RMSE while maintaining comparable inference speed.
> | Channel | 1 day | 3 day | 5 day | 7 day |
> |-|-|-|-|-|
> | t_850 | 0.83 | 1.56 | 2.48 | 3.32 |
> |z_500 | 86.67 | 253.34 | 486.38 | 693.32 |
> | t2m | 0.94 | 1.40 | 2.01 | 2.55 |
> | u_10 | 0.74 | 2.10 | 3.38 | 4.23 |
> | v_10 | 1.04 | 2.31 | 3.51 | 4.35 |
>
> **Response to weakness 3:**
> Generalization to extreme events: Section 4.6 demonstrates GaussianCast’s capability in predicting **tropical cyclones**, a representative extreme weather scenario. Our results show competitive performance against SOTA methods, indicating that GaussianCast can generalize to unseen extreme events.
> Effect of sampling rates on smoothness and local detail: Section 4.5 provides visualizations of reconstructed atmospheric fields at **different Gaussian sampling rates**. Further quantitative and qualitative evaluations are presented in the Appendix (Figures 8–15), which clearly illustrate how **varying sampling densities affect data smoothness and local detail preservation**.
>
> **Response to question 1:**
> Indeed, AIFS (Lang et al., 2024) employs a Reduced Gaussian Grid (RGG) with 542,080 points, which mitigates redundancy in latitude-longitude grids. However, AIFS still forecasts in grid space (albeit on an RGG), whereas our method uses the RGG only to initialize Gaussian positions and reconstructs the atmosphere as a continuous 3D Gaussian representation. Forecasting occurs in this 3D Gaussian space rather than on a 2D grid, providing:
> Data compression: The global atmosphere with only **69,185 continuous Gaussians**, far fewer than AIFS’s **542,080 RGG points**.
> Computational efficiency: Operating in the continuous 3D Gaussian space reduces redundancy and enables more efficient learning and inference.
> Conceptual novelty: Our approach predicts in a latent 3D Gaussian manifold. **By reformulating discrete grid-based forecasting into a continuous, resolution-agnostic generative process, GaussianCast enables reconstructions and forecasts at arbitrary spatial resolutions without retraining, a capability fundamentally absent in conventional grid-based models.**
>
> **Response to question 2:**
> Traditional 3DGS optimizes Gaussian parameters **per sample, minimizing reconstruction error for that specific atmospheric sample.** This leads to overfitting, as each Gaussian set essentially memorizes a single sample.
> GaussianCast overcomes this by adopting a conditional generative approach. Instead of optimizing Gaussians individually, the model **learns a mapping from the input atmospheric state to the Gaussian parameters. The model generalizes by learning the conditional distribution 𝑝(Gaussians∣atmospheric state), akin to conditional generative models in machine learning (e.g., VAE).**
> The theoretical basis is that conditioning on context **transforms a per-sample optimization problem into a learning problem over a distribution, enabling generalization**. In other words, the model learns how to generate Gaussians distribution from the atmospheric state, capturing the underlying data manifold rather than individual point instances.

---

> ### Author Response · Authors · 2025-11-22
>
> Dear Reviewer zvGn,
>
> Thank you for your detailed and insightful feedback on our paper.
>
> In our rebuttal, we addressed your concerns comprehensively. We have updated the visualization results and included them in the appendix section of the revised manuscript, as shown in Figures 8, 9, and 10. These figures visualizes the model’s weather forecasts on a global scale.
>
> We hope these updates have addressed your concerns, particularly regarding the clarity and completeness of the visual evidence. If any further clarification is needed, we would be happy to provide it. Otherwise, if you find our revisions satisfactory, we would be most grateful if you could consider revising your score upward.
>
> We sincerely value your time and expertise, and we look forward to any additional feedback you may have.
>
> Best regards,
>
> The Authors

---

### Meta-Review · Area_Chair_Mv2e · 2026-01-07

**Summary:**

The paper proposes to use 3d Gaussian Splatting (3DGS) for weather forecasting. The idea is to train a Graph Neural network (GNN)-based forecasting model that operates on a set of manually selected mesh nodes (a reduced Gaussian grid, RGG) that respect the spherical geometry of the globe. The model forecasts a 3D Gaussian covariance matrix, an "opacity" and a vector of predicted values per RGG point. To compute a forecast, an adaptive interpolation between neighboring RGG points is carried out, computing a weighted average of the predicted vectors at those points.

In addition to considering the criticism brought forward by the reviewers, I have read the paper myself. The paper shows strong empirical performance at a computational cost that is reported to be significantly lower than many alternatives. However, there are several issues with the paper and I believe that it is not yet ready for publication. The model is trained on a fixed RGG mesh, so the use of 3DGS appears to reduce to an adaptive (spatio-temporally varying) interpolation. This needs to be clarified and investigated in more detail, for instance analysing how much the "learned interpolation" differs between spatial positions and across time. One of the key promises of the paper is to enable forecasts at arbitrary spatial positions, but this is only evaluated in a small proof-of-concept that does not shed any light on whether the approach is actually effective or not.

The presentation could also be improved. For instance, it is not clear how the prediction in Eq 6 depends on the covariance matrix \Sigma.
The rationale for the 3DGS approach, which is the key methodological contribution, is not properly explained. I understand that you want to learn the Gaussian pdfs for the purpose of smoothly and adaptively interpolating the RGGs, but what it's the benefit of viewing this as a "rendering problem" rather than an adaptive interpolation problem? Specifically, in eq 6, what is the benefit of letting the Gaussians occlude each other instead of just computing a weighted sum of f-vectors, where the weights are given by values of the Gaussian pdfs at the query points. I don't claim that it's not a good idea, but the rationale for the idea in the context of weather forecasting should be explained more clearly.


Finally, there seems to be some inconsistencies between what is stated in the paper and in the rebuttal regarding computational cost. The paper states a training time of 10 days, whereas the rebuttal mentions 3 days and I could not see any explanation of the difference, which makes me question these results.

**Reviewer Concerns:**

I found the reviews to be of rather low quality. Reviewer ZQTg questioned the motivation to speed up MLWP models further, which I think is properly addressed in the rebuttal, but there are some inconsistencies in the numbers reported in the rebuttal and the paper (see meta review)

**Reviewer Scores:**

N/A

---

### Decision · Program_Chairs · 2026-01-26

Reject